# On the Identifiability of Sparse ICA without Assuming Non-Gaussianity

**Ignavier Ng**[*1], **Yujia Zheng**[*1], **Xinshuai Dong**[1], **Kun Zhang**[1,2]

[1] Carnegie Mellon University
[2] Mohamed bin Zayed University of Artificial Intelligence
{ignavierng, yujiazh, dongxinshuai, kunz1}@cmu.edu

## Abstract

Independent component analysis (ICA) is a fundamental statistical tool used to reveal hidden generative processes from observed data. However, traditional ICA approaches struggle with the rotational invariance inherent in Gaussian distributions, often necessitating the assumption of non-Gaussianity in the underlying sources. This may limit their applicability in broader contexts. To accommodate Gaussian sources, we develop an identifiability theory that relies on second-order statistics without imposing further preconditions on the distribution of sources, by introducing novel assumptions on the connective structure from sources to observed variables. Different from recent work that focuses on potentially restrictive connective structures, our proposed assumption of structural variability is both considerably less restrictive and provably necessary. Furthermore, we propose two estimation methods based on second-order statistics and sparsity constraint. Experimental results are provided to validate our identifiability theory and estimation methods.

## 1 Introduction

Independent component analysis (ICA) [12] has emerged as an essential statistical tool in the scientific community, with application in various disciplines including neuroscience [27], biology [41, 9], and Earth science [32]. It aims to uncover the hidden generative processes that govern the observed data and separate mixed signals into independent sources. However, it is known that the traditional approaches of ICA struggle with Gaussian sources, which may limit their applicability in a wide variety of contexts. For instance, several biological traits and measurements such as height, blood pressure, and measurement errors in genetics, as well as the thermal noises in electronic circuits, may often be normally distributed [29, 34]. Classical identifiability results in ICA rely on higher-order statistics (e.g., Kurtois) [24], and cannot provide desired theoretical guarantees when there is more than one Gaussian source. Identifiability based on second-order statistic is thus essential in these scenarios.

The primary hurdle in applying ICA to Gaussian sources lies in the rotational invariance of Gaussian distribution [25]. To address this issue, earlier studies [30, 5, 35] incorporated additional information by assuming that the sources are nonstationary. However, this extra information may not always be readily available, limiting the generalizability of these approaches. Thus, recent research [49, 1] has started to delve into the connective structure between the sources and the observed variables, as opposed to solely focusing on distributional assumptions (e.g., non-Gaussianity and nonstationarity). This shift of focus is motivated by a key observation: despite the rotational invariance in the distribution of Gaussian sources, the sparsity of mixing matrix undergoes noticeable changes, i.e., it may be denser after rotation [49]. Building on this insight, Zheng et al. [49] introduced two assumptions on the support of the mixing matrix to achieve identifiability of Gaussian sources, leading to a novel perspective for tackling this long-standing challenge in the field of ICA. On the other hand, Abrahamsen and Rigollet [1] assumed that the mixing matrix is generated from a sparse Bernoulli-Gaussian ensemble.

---

[*]Equal contribution.

37th Conference on Neural Information Processing Systems (NeurIPS 2023).

Although the rotational invariance of Gaussian distribution can be resolved by the structural assumptions on the mixing matrix proposed by Zheng et al. [49], they may be deemed overly restrictive. For instance, both of their structural assumptions cannot deal with the case where the set of observed variables influenced by one source is a subset of those affected by another source, which may not be uncommon in practice. This may limit the applicability of ICA in complex real-world scenarios, thus underscoring the need for a weaker and more flexible structural assumption that is capable of addressing the rotational invariance in a more universally applicable manner.

To enhance the applicability with Gaussian sources, we develop an identifiability theory of ICA from second-order statistics under more flexible structural constraints. We introduce a novel assumption, namely *structural variability*, that is considerably weaker than existing ones. Notably, this assumption is proved to be among the necessary conditions for identifying Gaussian sources by focusing on the connective structure (i.e., the support of the mixing matrix). Moreover, we propose two estimation methods grounded in sparsity regularization and continuous constrained optimization. The efficacy of our proposed methods has been validated through experiments, which also reaffirm the validity of our theoretical result. Lastly, as a matter of independent interest, we establish the connection between our identifiability result of ICA with causal discovery from second-order statistics; our finding further bridges the gap between these two fields and provides insights into the interpretation of our result.

## 2 Problem Setting

We consider the ICA setup given by $\mathbf{x} = \tilde{\mathbf{A}}\mathbf{s}$, where $\mathbf{x} \in \mathbb{R}^n$ denotes the observed random vector, $\mathbf{s} \in \mathbb{R}^n$ is the latent random vector representing the independent components, also called sources, and $\tilde{\mathbf{A}} = [\tilde{\mathbf{a}}_1 | \cdots | \tilde{\mathbf{a}}_n] \in \mathbb{R}^{n \times n}$ denotes the unknown mixing matrix. We assume here that $\tilde{\mathbf{A}}$ has full rank and all sources are standardized. For the estimated mixing matrix $\hat{\mathbf{A}}$ and the ground-truth one $\tilde{\mathbf{A}}$, we write $\hat{\mathbf{A}} \sim \tilde{\mathbf{A}}$ if they differ only in column permutations and sign changes of columns, and $\hat{\mathbf{A}} \not\sim \tilde{\mathbf{A}}$ vice versa. The goal is then to estimate $\hat{\mathbf{A}}$ such that $\hat{\mathbf{A}} \sim \tilde{\mathbf{A}}$; in this case, one could identify a one-on-one mapping between the ground truth sources and the estimated ones, i.e., the unknown mixing process has been demixed during estimation. Since the support of a mixing matrix $\mathbf{A}$ essentially represents the connective structure between sources and observed variables, we have the following definition for the ease of reference.

**Definition 1 (Connective Structure).** *Given a mixing matrix $\mathbf{A}$, we define its connective structure as a directed bipartite graph $\mathcal{G}_{\mathbf{A}} = (\mathcal{V}_{\mathbf{A}}, \mathcal{E}_{\mathbf{A}})$ from sources $\mathbf{s}$ to observed variables $\mathbf{x}$, where the nodes and edges are defined as $\mathcal{V}_{\mathbf{A}} := \{s_i\}_{i=1}^n \cup \{x_i\}_{i=1}^n$ and $\mathcal{E}_{\mathbf{A}} := \{(s_j, x_i) : a_{ij} \neq 0\}$, respectively.*

**Notations.** We use bold capital letters (e.g., $\mathbf{A}$), bold lowercase letters (e.g., $\mathbf{a}$), and italic letters (e.g., $a$) to denote matrices, vectors, and scalar quantities, respectively. For any matrix $\mathbf{A}$, we denote its $j$-th column by $\mathbf{a}_j$, $i$-th row by $\mathbf{a}_{i,:}$, and $(i,j)$-th entry by $a_{i,j}$. We also denote by $\mathbf{A}_{\mathcal{J}}$ the submatrix of $\mathbf{A}$ by obtaining the columns indexed by set $\mathcal{J}$. For any vector $\mathbf{a}$, we denote its $i$-th entry by $a_i$. We define the support set of matrix $\mathbf{A}$ as $\operatorname{supp}(\mathbf{A}) := \{(i,j) : a_{i,j} \neq 0\}$, and its support matrix as $\boldsymbol{\xi}_{\mathbf{A}}$ which is of the same size as $\mathbf{A}$, where $(\boldsymbol{\xi}_{\mathbf{A}})_{i,j} = \times$ if $a_{i,j} \neq 0$ and $(\boldsymbol{\xi}_{\mathbf{A}})_{i,j} = 0$ otherwise. The notations of support set and support matrix are similarly defined for vector $\mathbf{a}$. We denote by $\|\mathbf{A}\|_0$ the number of nonzero entries in $\mathbf{A}$, and we have $\|\mathbf{A}\|_0 = |\operatorname{supp}(\mathbf{A})| = \|\boldsymbol{\xi}_{\mathbf{A}}\|_0$. Furthermore, we denote the $n \times n$ identity matrix and $m \times n$ zero matrix by $\mathbf{I}_n$ and $\mathbf{0}_{m \times n}$, respectively; to lighten the notation, we drop their subscripts when the context is clear. We also use $[n]$ to denote $\{1, 2, \ldots, n\}$.

## 3 Identifiability Result without Assuming Non-Gaussianity

By exploiting the non-Gaussianity of the sources, such as fourth-order cumulant, existing approaches are able to estimate the true mixing matrix $\tilde{\mathbf{A}}$ up to signed column permutation when there is at most one Gaussian source [23, 2]. However, these approaches typically fail in the presence of more than one Gaussian source, because higher-order statistics cannot be utilized for full identifiability. The primary challenge of achieving identifiability for Gaussian sources lies in the rotational invariance of the Gaussian distribution. More specifically, the second-order statistics (or, more specifically, population-level covariance matrix) $\tilde{\boldsymbol{\Sigma}} = \tilde{\mathbf{A}}\tilde{\mathbf{A}}^\top$ remains unchanged if one replaces $\tilde{\mathbf{A}}$ with $\tilde{\mathbf{A}}\mathbf{U}$ for any orthogonal matrix $\mathbf{U}$. Therefore, considering only second-order statistics, the true mixing matrix $\tilde{\mathbf{A}}$ is, generally speaking, only identifiable up to right orthogonal transformation without

further assumptions. In this section, we adopt a different perspective that departs from traditional distributional assumptions (i.e., non-Gaussianity and fourth-order cumulant), and instead introduces novel and precise assumptions on the connective structure, specifically the support of the mixing matrix. These assumptions enable identification of the true mixing matrix $\tilde{\mathbf{A}}$ up to signed column permutation using second-order statistics and sparsity constraint. Roughly speaking, with the population-level covariance matrix $\tilde{\boldsymbol{\Sigma}}$, we consider the following formulation:

$$\min_{\mathbf{A} \in \mathbb{R}^{n \times n}} \|\mathbf{A}\|_0 \quad \text{subject to} \quad \mathbf{A}\mathbf{A}^\top = \tilde{\boldsymbol{\Sigma}} = \tilde{\mathbf{A}}\tilde{\mathbf{A}}^\top. \tag{1}$$

Note that we start with the assumption of $\mathbf{A}\mathbf{A}^\top = \tilde{\boldsymbol{\Sigma}}$ in the formulation above and our identifiability result (e.g., in Theorem 1); such an assumption can be obtained, e.g., from the equality of Gaussian likelihoods in the large sample limit. This also inspires our sparsity-regularized likelihood-based estimation method that will be described in Section 4.2, which is more inline with, e.g., model selection approaches based on sparsity-regularized likelihood [36, 18].

In Section 3.1, we first examine various types of constraints arising from second-order statistics. We then present the main identifiability result in Section 3.2. We show that our assumptions are strictly weaker than existing sparsity assumptions in Section 3.3, and establish the connection between our identifiability theory with causal discovery in Section 3.4. All proofs are given in Appendices C and D.

## 3.1 Semialgebraic Constraints Arising from Second-Order Statistics

We first discuss various notions related to constraints arising from covariance matrices of observed variables $\mathbf{x}$, which serve as a fundamental basis for introducing our assumptions and identifiability result of ICA in Section 3.2. These notions have been studied in the field of algebraic statistics [16], factor analysis [15], graphical models [19, 40], and causality [14, 20].

We begin with the following definition on the set of covariance matrices entailed by $\boldsymbol{\xi}$ for different values of the free parameters in $\boldsymbol{\xi}$.

**Definition 2 (Covariance Set).** *The covariance set of support matrix $\boldsymbol{\xi}$ is defined as*

$$\boldsymbol{\Sigma}(\boldsymbol{\xi}) := \{\mathbf{A}\mathbf{A}^\top : \mathbf{A} \in \mathbb{R}^{n \times n}, \operatorname{supp}(\mathbf{A}) \subseteq \operatorname{supp}(\boldsymbol{\xi}), \mathbf{A} \text{ is non-singular}\}.$$

The support $\boldsymbol{\xi}$ of mixing matrix $\mathbf{A}$ imposes certain constraints on the entries of the covariance matrix $\boldsymbol{\Sigma} = \mathbf{A}\mathbf{A}^\top$, which, by Tarski–Seidenberg theorem (see [6]), correspond to *semialgebraic constraints*, i.e., equality and inequality constraints. The covariance set $\boldsymbol{\Sigma}(\boldsymbol{\xi})$ is then said to be a *semialgebraic set*, i.e., a set that can be described with a finite number of polynomial equations and inequalities [6]. Clearly, if a covariance matrix $\boldsymbol{\Sigma}$ belongs to the covariance set $\boldsymbol{\Sigma}(\boldsymbol{\xi})$, then $\boldsymbol{\Sigma}$ satisfies the semialgebraic constraints imposed by $\boldsymbol{\xi}$.

For an equality constraint, the set of values satisfying the constraint has zero Lebesgue measure over the parameter space involved. Given a support matrix $\boldsymbol{\xi}$, we denote by $H(\boldsymbol{\xi})$ the set of equality constraints it imposes on the corresponding covariance matrices. On the other hand, the set of values satisfying an inequality constraint has nonzero Lebesgue measure. To illustrate these constraints, we provide a three-variable example below. Note that the example only serves as illustrations of the constraints; our estimation methods (in Section 4) do not require deriving them in practice.

**Example 1 (Semialgebraic Constraints).** *Consider support matrices*

$$\boldsymbol{\xi}_1 = \begin{bmatrix} \times & 0 & 0 \\ \times & \times & 0 \\ \times & 0 & \times \end{bmatrix} \quad \text{and} \quad \boldsymbol{\xi}_2 = \begin{bmatrix} \times & 0 & \times \\ \times & \times & 0 \\ 0 & \times & \times \end{bmatrix}.$$

*The equality constraints imposed by $\boldsymbol{\xi}_1$ include*

$$\Sigma_{1,1}\Sigma_{2,3} - \Sigma_{1,2}\Sigma_{1,3} = 0,$$

*while the inequality constraints imposed by $\boldsymbol{\xi}_2$ include*

$$(\Sigma_{1,1}\Sigma_{2,2}\Sigma_{3,3} + \Sigma_{1,1}\Sigma_{2,3}^2 - \Sigma_{2,2}\Sigma_{1,3}^2 - \Sigma_{3,3}\Sigma_{1,2}^2)^2 - 4(\Sigma_{1,1}\Sigma_{2,2} - \Sigma_{1,2}^2)(\Sigma_{1,1}\Sigma_{3,3}\Sigma_{2,3}^2 - \Sigma_{1,3}^2\Sigma_{2,3}^2) \geq 0.$$

The detailed derivation can be found in Appendix D.1 and provides insights into how such constraints arise from the corresponding support matrices. In the example above, the covariance matrix $\boldsymbol{\Sigma}$ generated by any mixing matrix $\mathbf{A}$ with support $\boldsymbol{\xi}_1$ must satisfy the corresponding equality constraint; similarly, the covariance matrix $\boldsymbol{\Sigma}$ generated by any mixing matrix $\mathbf{A}$ with support $\boldsymbol{\xi}_2$ must satisfy the above inequality constraint. These constraints serve as footprints of the mixing matrix on the covariance matrix, and can be exploited for its identifiability, which we explain in the next section.

## 3.2 Identifiability Result from Second-Order Statistics

In this section, we present our identifiability result of ICA from second-order statistics. The core idea is to introduce precise and mild assumptions on the connective structure from sources to observed variables. These assumptions facilitate the identification of the mixing matrix through the application of a sparsity constraint, formulated in Problem (1). To begin, we describe our primary assumption concerning the connective structure as follows.

**Assumption 1 (Structural Variability).** *Every pair of the columns in the support matrix of* $\mathbf{A}$ *differ in more than one entry. That is, for every* $i, j \in [n]$ *and* $i \neq j$*, we have*

$$| \operatorname{supp}(\mathbf{a}_i) \cup \operatorname{supp}(\mathbf{a}_j)| - | \operatorname{supp}(\mathbf{a}_i) \cap \operatorname{supp}(\mathbf{a}_j)| > 1.$$

The assumption above implies that every pair of sources should influence more than one different observed variable. Notably, in the field of nonlinear ICA with auxiliary variable, Hyvärinen and Morioka [22], Hyvärinen et al. [26] have adopted the assumption of *sufficient variability* which requires that the auxiliary variable has a sufficiently diverse effect on the distributions of sources; specifically, the conditional distributions of the sources given the auxiliary variable must vary sufficiently. In contrast, our assumption of *structural variability* requires that every pair of sources influence sufficiently diverse sets of observed variables, facilitating the disentanglement of each source.

We provide several examples in Appendix E.1 to illustrate the broad applicability of the assumption above. Furthermore, the following proposition justifies such an assumption because it is a necessary condition for identifiability via second-order statistics and sparsity. The intuition is that if Assumption 1 is violated, there exists a rotation that maps matrix $\tilde{\mathbf{A}}$ to another matrix $\hat{\mathbf{A}}$ which has equal or smaller number of nonzero entries and is not a column permutation of $\tilde{\mathbf{A}}$.

**Proposition 1.** *If the true mixing matrix* $\tilde{\mathbf{A}}$ *does not satisfy Assumption 1, then there exists a solution* $\hat{\mathbf{A}}$ *to Problem* (1) *such that* $\hat{\mathbf{A}} \not\sim \tilde{\mathbf{A}}$*.*

**Remark 1 (Necessary Condition).** *Assumption 1 is a necessary condition for identifiability of ICA via second-order statistics and under sparsity constraint.*

We also adopt the following assumption on the mixing matrix for the identifiability of ICA.

**Assumption 2 (Permutations to Lower Triangular Matrix).** *The matrix* $\mathbf{A}$ *can be permuted by independent row and column permutations to be lower triangular. That is, there exist permutation matrices* $\mathbf{P}_1$ *and* $\mathbf{P}_2$ *such that* $\mathbf{P}_1^\top \mathbf{A} \mathbf{P}_2$ *is lower triangular.*

As we show in the proof of identifiability result in Theorem 1, Assumption 2, loosely speaking, ensures that the resulting covariance matrix does not contain "nontrivial" inequality constraints. In Example 1, support matrix $\boldsymbol{\xi}_1$ satisfies Assumption 2 and leads to an equality constraint, while matrix $\boldsymbol{\xi}_2$ fails to meet this assumption, resulting in an inequality constraint. The Lebesgue measure of the parameters leading to such inequality constraint is not zero, thus requiring additional assumptions to handle such cases. Therefore, we adopt Assumption 2 in this work and focus on equality constraints.

A key ingredient of our identifiability result based on sparsity is the dimension of the covariance set $\boldsymbol{\Sigma}(\boldsymbol{\xi})$. It may be natural to expect that the dimension of $\boldsymbol{\Sigma}(\boldsymbol{\xi})$, denoted as $\dim(\boldsymbol{\Sigma}(\boldsymbol{\xi}))$, equals the number of parameters used to specify the mixing matrices, i.e., $\|\boldsymbol{\xi}\|_0$. This is not the case for general mixing matrices, but we show that such property holds under Assumption 2.

**Proposition 2 (Dimension of Covariance Set).** *Let* $\boldsymbol{\xi}$ *be a support matrix that satisfies Assumption 2. Then, its covariance set has a dimension of* $\|\boldsymbol{\xi}\|_0$*, i.e.,* $\dim(\boldsymbol{\Sigma}(\boldsymbol{\xi})) = \|\boldsymbol{\xi}\|_0$*.*

Note that Assumption 2 allows independent row and column permutations, which thus may be rather mild especially for sparse mixing matrix. Below we provide an example of the connective structure that satisfies this assumption. We also introduce an efficient approach to verify whether a mixing matrix satisfies Assumption 2 in Appendix E.2.

**Example 2.** *If the connective structure* $\mathcal{G}_{\mathbf{A}}$ *of mixing matrix* $\mathbf{A}$ *is a polytree, then matrix* $\mathbf{A}$ *satisfies Assumption 2.*

Finally, the following assumption is needed to ensure that the equality constraints arising from the covariance matrix are entailed by the true mixing matrix, rather than accidental parameter cancellations. This establishes a correspondence between equality constraints in the covariance matrix and those imposed by the support of the mixing matrix. Similar assumption has been employed in various tasks such as causal discovery [39, 20], as discussed in Section 3.4.

**Assumption 3 (Faithfulness).** *For a mixing matrix $\mathbf{A}$ and the resulting covariance matrix $\boldsymbol{\Sigma}$, $\boldsymbol{\Sigma}$ satisfies an equality constraint $\kappa$ only if $\kappa \in H(\boldsymbol{\xi}_{\mathbf{A}})$.*

The following proposition justifies Assumption 3 and demonstrates that it is a generic property in the sense that it holds almost everywhere in the space of possible mixing matrices. In other words, it is only violated for a set of mixing matrices with zero Lebesgue measure.

**Proposition 3 (Generic Property).** *Suppose that the nonzero coefficients of matrix $\mathbf{A}$ are randomly drawn from a distribution that is absolutely continuous with respect to Lebesgue measure. Then, matrix $\mathbf{A}$ satisfies Assumption 3 with probability one.*

With the aforementioned assumptions in place, we are ready to present our main identifiability result of ICA: by making use of second-order statistics and sparsity constraint, we show that the true mixing matrix can be identified up to signed column permutation. The key intuition is as follows: although in general there is rotational invariance in second-order statistics, the sparsity of the estimated mixing matrix, under our assumptions above, is guaranteed to be denser than the true mixing matrix after any rotation (except permutation). Note that the proof leverages the technical tools developed by Ghassami et al. [20] which involve different types of rotations.

**Theorem 1 (Identifiability with Sparsity).** *Suppose that the true mixing matrix $\tilde{\mathbf{A}}$ satisfies Assumptions 1, 2, and 3. Let $\hat{\mathbf{A}}$ be a solution of the following problem:*

$$\min_{\mathbf{A} \in \mathbb{R}^{n \times n}} \|\mathbf{A}\|_0 \quad \text{subject to} \quad \mathbf{A}\mathbf{A}^\top = \tilde{\mathbf{A}}\tilde{\mathbf{A}}^\top \quad \text{and} \quad \mathbf{A} \text{ satisfies Assumption 2.} \tag{2}$$

*Then, we have $\hat{\mathbf{A}} \sim \tilde{\mathbf{A}}$.*

## 3.3 Related Sparsity Assumptions

As briefly discussed in Section 1, Zheng et al. [49] have also provided identifiability result of ICA without assuming non-Gaussianity. Their approach, similar to ours, relies on assumptions related to the support of the mixing matrix, which are provided below.

**Assumption 4 (Zheng et al. [49]).** *For every set $\mathcal{I} \subseteq [n]$ where $|\mathcal{I}| > 1$ and for all $i \in \mathcal{I}$, we have*

$$\left| \bigcup_{j \in \mathcal{I}} \text{supp}(\mathbf{a}_j) \right| - \text{rank}(\text{overlap}(\mathbf{A}_{\mathcal{I}})) > |\text{supp}(\mathbf{a}_i)|. \tag{3}$$

**Assumption 5 (Zheng et al. [49]).** *For every $i \in [n]$, there exists set $\mathcal{I} \subset [n]$ such that*

$$\bigcap_{j \in \mathcal{I}} \text{supp}(\mathbf{a}_{j,:}) = \{i\}.$$

While Assumptions 4 and 5 shed light on resolving the long-standing challenge of the rotational indeterminacy of Gaussian sources, their general applicability remains unclear. One notable restriction is that they require each column of the support matrix to not be a subset of the other, formalized below. This may be overly restrictive in practical scenarios and is not the case for our Assumption 1.

**Assumption 6 (Column Subset).** *Each column in the support of $\mathbf{A}$ is not a subset of the other.*

Since the true generating process of real-world data is inaccessible, it is challenging to quantitatively evaluate the applicability of these sparsity assumptions. In light of this challenge, we demonstrate the significance and advantage of our Assumption 1 by proving that it is strictly weaker than Assumptions 4 and 5. This also strengthens the validity of our result (i.e., Proposition 1) that Assumption 1 is a necessary condition for achieving identifiability with second-order statistics and sparsity constraint.

**Theorem 2.** *For mixing matrix $\mathbf{A}$, we have the following chain of chain of implications:*

$$\textit{Assumption 4} \implies \textit{Assumption 6} \implies \textit{Assumption 1}.$$

*Furthermore, there exists a matrix $\mathbf{A}$ satisfying Assumption 1 that does not satisfy Assumption 4.*

**Theorem 3.** *For mixing matrix $\mathbf{A}$, we have the following chain of chain of implications:*

$$\textit{Assumption 5} \implies \textit{Assumption 6} \implies \textit{Assumption 1}.$$

*Furthermore, there exists a matrix $\mathbf{A}$ satisfying Assumption 1 that does not satisfy Assumption 5.*

**Remark 2.** *Assumption 1 is not only strictly weaker than both Assumption 4 and 5, but also accommodates cases where one column of mixing matrix $\mathbf{A}$ is a subset of of another (provided that their supports differ in more than one entry). On the other hand, the latter two assumptions cannot accommodate such cases, further demonstrating the flexibility and broader scope of Assumption 1.*

## 3.4 Connection with Causal Discovery

ICA has emerged as a useful tool for causal discovery over the past two decades [38]. In particular, Shimizu et al. [38] demonstrated that the identifiability of ICA based on non-Gaussianity can be leveraged to discover the complete structure of a linear non-Gaussian structural equation model (SEM). In this section, we establish the connection and provide an analogy between ICA and causal discovery from second-order statistics. This connection further bridges the gap between these two fields, and provides insights into the interpretation of our identifiability result.

Let $\mathbb{R}_{\text{off}}^{n \times n}$ be the set of matrices whose diagonal entries are zero, and $\text{diag}(\mathbb{R}_{>0}^n)$ be the set of positive diagonal matrices. Consider the linear SEM $\mathbf{x} = \tilde{\mathbf{B}}^\top \mathbf{x} + \mathbf{e}$, where $\mathbf{x}$ denotes the random vector, $\tilde{\mathbf{B}} \in \mathbb{R}_{\text{off}}^{n \times n}$ denotes the weighted adjacency matrix representing a directed graph without self-loop, and $\mathbf{e}$ is the independent noise vector with covariance matrix $\tilde{\mathbf{\Omega}} \in \text{diag}(\mathbb{R}_{>0}^n)$. The graph is often assumed to be a directed acyclic graph (DAG); in this case, two DAGs are said to be *Markov equivalent* if they share the same skeleton and v-structures [43], resulting in the same set of conditional independencies. Also, the inverse covariance matrix of $\mathbf{x}$ is given by $\tilde{\mathbf{\Theta}} = (\mathbf{I} - \tilde{\mathbf{B}})\tilde{\mathbf{\Omega}}^{-1}(\mathbf{I} - \tilde{\mathbf{B}})^\top$. We refer readers to Spirtes et al. [39], Glymour et al. [21] for more details and a review of causal discovery.

Score-based method is a major class of causal discovery methods that optimizes a goodness-of-fit measure under a sparsity constraint [21], e.g., BIC [36]. In essence, score-based causal discovery from second-order statistics can often be formulated in the large sample limit as the following optimization problem (the commonly used acyclicity constraint is omitted here and will be clarified subsequently):

$$\min_{\substack{\mathbf{B} \in \mathbb{R}_{\text{off}}^{n \times n}, \\ \mathbf{\Omega} \in \text{diag}(\mathbb{R}_{>0}^n)}} \|\mathbf{B}\|_0 \quad \text{subject to} \quad (\mathbf{I} - \mathbf{B})\mathbf{\Omega}^{-1}(\mathbf{I} - \mathbf{B})^\top = \tilde{\mathbf{\Theta}} = (\mathbf{I} - \tilde{\mathbf{B}})\tilde{\mathbf{\Omega}}^{-1}(\mathbf{I} - \tilde{\mathbf{B}})^\top. \quad (4)$$

By substituting $\mathbf{A} := (\mathbf{I} - \mathbf{B})\mathbf{\Omega}^{-\frac{1}{2}}$ into the above formulation, we obtain the ICA formulation with second-order statistics and sparsity constraint introduced in Problem (1). To establish a precise connection between formulations (4) and (1), we present the following theorem which indicates that these formulations can be translated into each other.

**Theorem 4 (Equivalent Formulations).** *Suppose $\tilde{\mathbf{A}} = (\mathbf{I} - \tilde{\mathbf{B}})\tilde{\mathbf{\Omega}}^{-\frac{1}{2}}$. Then, we have:*

(a) *Let $(\hat{\mathbf{B}}, \hat{\mathbf{\Omega}})$ be a solution to Problem (4). Then, $\hat{\mathbf{A}} := (\mathbf{I} - \hat{\mathbf{B}})\hat{\mathbf{\Omega}}^{-\frac{1}{2}}$ is a solution to Problem (1).*

(b) *Let $\hat{\mathbf{A}}$ be a solution to Problem (1). Then, there exist matrices $\hat{\mathbf{B}} \in \mathbb{R}_{\text{off}}^{n \times n}$ and $\hat{\mathbf{\Omega}} \in \text{diag}(\mathbb{R}_{>0}^n)$ such that $\hat{\mathbf{A}} \sim (\mathbf{I} - \hat{\mathbf{B}})\hat{\mathbf{\Omega}}^{-\frac{1}{2}}$, and $(\hat{\mathbf{B}}, \hat{\mathbf{\Omega}})$ is a solution to Problem (4).*

Thus, the formulations of causal discovery and ICA via second-order statistics and sparsity constraint share inherent similarities. The key difference lies in their respective goals–the former aims to estimate the support of $\tilde{\mathbf{B}}$ up to a Markov equivalence class [39], while the latter aims to estimate $\tilde{\mathbf{A}}$ up to signed column permutation. The other difference is that $(\mathbf{I} - \tilde{\mathbf{B}})\tilde{\mathbf{\Omega}}^{-1}(\mathbf{I} - \tilde{\mathbf{B}})^\top$ represents the inverse covariance matrix $\tilde{\mathbf{\Theta}}$ of $\mathbf{x}$ in causal discovery, while $\tilde{\mathbf{A}}\tilde{\mathbf{A}}^\top$ represents the covariance matrix $\tilde{\mathbf{\Sigma}}$ of $\mathbf{x}$ in ICA.

In addition to establishing the connection between formulations (4) and (1), we show that the assumptions we employ for identifiability of ICA, namely Assumptions 1, 2, and 3, are inherently related to causal discovery. Notably, Assumption 3 has been used in causal discovery [39, 20] to ensure that the conditional independencies in the distribution are entailed by the true directed graph. We now present a result that establishes the connection of Assumptions 1 and 2 with causal discovery.

**Theorem 5.** *Suppose $\mathbf{A} \sim (\mathbf{I} - \mathbf{B})\mathbf{\Omega}^{-\frac{1}{2}}$ for matrices $\mathbf{A} \in \mathbb{R}^{n \times n}$, $\mathbf{B} \in \mathbb{R}_{\text{off}}^{n \times n}$, and $\mathbf{\Omega} \in \text{diag}(\mathbb{R}_{>0}^n)$. Then, $\mathbf{A}$ satisfies Assumptions 1 and 2 if and only if $\mathbf{B}$ represents a DAG whose Markov equivalence class is a singleton.*

In causal discovery, it is rather common to assume that the true directed graph is acyclic and accordingly incorporate an acyclicity constraint to formulation (4). As indicated in Theorem 5 (and Proposition 10 in Appendix D.8), this acyclicity assumption corresponds to Assumption 2 in the context of ICA. Therefore, Theorem 4 can be straightforwardly extended to show the equivalence between formulations (2) and (4) with an additional acyclicity constraint on matrix $\mathbf{B}$. Furthermore, it is worth noting that the mapping from mixing matrix $\mathbf{A}$ satisfying Assumption 2 to a DAG is unique, which is straightforwardly implied by Shimizu et al. [38, Appendix A].

**Proposition 4 (Shimizu et al. [38]).** *Suppose matrix $\mathbf{A}$ is non-singular and satisfies Assumption 2. Then, there exist unique matrices $\mathbf{B} \in \mathbb{R}_{\text{off}}^{n \times n}$ and $\boldsymbol{\Omega} \in \text{diag}(\mathbb{R}_{>0}^n)$ such that $\mathbf{A} \sim (\mathbf{I} - \mathbf{B})\boldsymbol{\Omega}^{-\frac{1}{2}}$. Furthermore, matrix $\mathbf{B}$ represents a DAG.*

Moreover, as indicated in Theorem 5 (and Proposition 11 in Appendix D.8), Assumption 1 implies that the Markov equivalence class of $\mathbf{B}$ is a singleton; in this case, the true DAG can be completely identified. In particular, the Markov equivalence class of DAG is a singleton when all edges are either part of a v-structure or required to be oriented to avoid forming new v-structures or cycles [31, 4].

## 4   Estimation Methods with Second-Order Statistics

Building upon the identifiability result provided in Section 3, we propose two estimation methods that leverage second-order statistics and sparsity. These methods involve solving a continuous constrained optimization problem, which we discuss in detail in this section. First, in Section 4.1, we introduce a novel approach to formulate the search space in Problem (2) that enables the application of continuous optimization techniques. We then describe the proposed estimation methods in Section 4.2. All proofs are provided in Appendix D.

### 4.1   Characterization of Search Space

The key to achieving the identifiability result presented in Theorem (1) lies in the optimization problem (2), where the search space involves the matrices $\mathbf{A}$ that satisfy Assumption 2. Consequently, a crucial question arises: is there an efficient approach for exploring the space of matrices $\mathbf{A}$ that satisfy Assumption 2? Inspired by Zheng et al. [48], Wei et al. [45], Zhang et al. [47], we introduce the following function to characterize the search space:

$$g(\mathbf{A}) = \text{tr}\left(\sum_{k=2}^n (\text{off}(\mathbf{A}) \odot \text{off}(\mathbf{A}))^k\right), \quad \text{where} \quad (\text{off}(\mathbf{A}))_{i,j} = \begin{cases} 0, & \text{if } i = j, \\ a_{i,j}, & \text{otherwise.} \end{cases}$$

Here, symbol $\odot$ denotes the Hadamard product. We then provide the following lemma that establishes the relationship between function $g(\mathbf{A})$ and a specific type of permutation, namely simultaneous equal row and column permutation.

**Lemma 1.** *For any matrix $\mathbf{A}$, $g(\mathbf{A}) = 0$ if and only if it can be permuted via simultaneous equal row and column permutations to be lower triangular.*

Intuitively speaking, if we interpret matrix $\mathbf{A}$ as a weighted adjacency matrix of a directed graph, say $\mathcal{G}$, then $\text{tr}((\text{off}(\mathbf{A}) \odot \text{off}(\mathbf{A}))^k)$ counts the number of length-$k$ weighted closed walks in $\mathcal{G}$ excluding the self-loops. Therefore, $g(\mathbf{A})$ counts the total number of weighted closed walks in $\mathcal{G}$ without including self-loops. $g(\mathbf{A}) = 0$ then implies that $\mathcal{G}$ does not contain any cycle longer than one (i.e., it may contain self-loops). It is known that a directed graph is acyclic if and only if its weighted adjacency matrix can be permuted via simultaneous equal row and column permutations to be *strictly* lower triangular. In our case, $\mathcal{G}$ may contain self-loops, and thus can be permuted to a lower triangular form. This distinction elucidates the difference between our characterization and that introduced by Zheng et al. [48], Zhang et al. [47], i.e., their characterization focuses on matrices that are strictly lower triangular, while ours focuses on lower triangular matrices.

The following proposition sheds light on the connection between $g(\mathbf{A})$ and Assumption 2.

**Proposition 5.** *The matrix $\mathbf{A}$ satisfies Assumption 2 if and only if there is a matrix $\hat{\mathbf{A}}$ such that it is a column permutation of $\mathbf{A}$ and that $g(\hat{\mathbf{A}}) = 0$.*

The above proposition indicates that the search for matrices $\mathbf{A}$ satisfying Assumption 2 can be effectively conducted by considering the constraint $g(\mathbf{A}) = 0$. Accordingly, we establish an alternative formulation of the identifiability result presented in Section 3.2. In the following section, we will introduce efficient approaches for solving Problem (5).

**Theorem 6 (Alternative Formulation of Identifiability).** *Suppose that the true mixing matrix $\tilde{\mathbf{A}}$ satisfies Assumptions 1, 2, and 3. Let $\hat{\mathbf{A}}$ be a solution of the following problem:*

$$\min_{\mathbf{A} \in \mathbb{R}^{n \times n}} \|\mathbf{A}\|_0 \quad \text{subject to} \quad \mathbf{A}\mathbf{A}^\top = \tilde{\mathbf{A}}\tilde{\mathbf{A}}^\top \quad \text{and} \quad g(\mathbf{A}) = 0. \tag{5}$$

*Then, we have $\hat{\mathbf{A}} \sim \tilde{\mathbf{A}}$.*

---

**Algorithm 1** Decomposition-Based Method

---

**Require:** initial penalty coefficient $c_1 > 0$; multiplicative factor $\beta > 1$; maximum number of iterations $k_{\max} > 0$; tolerance $\epsilon_1, \epsilon_2 > 0$; initial solution $\mathbf{A}_0$; empirical covariance matrix $\bar{\boldsymbol{\Sigma}}$
1: **for** $k = 1, 2, \ldots, k_{\max}$ **do**
2:     Solve $\mathbf{A}_k := \arg\min_{\mathbf{A} \in \mathbb{R}^{n \times n}} \rho(\mathbf{A}) + \frac{c_k}{2}\|\mathbf{A}\mathbf{A}^\top - \bar{\boldsymbol{\Sigma}}\|_F^2 + \frac{c_k}{2}g(\mathbf{A})^2$ initialized at $\mathbf{A}_{k-1}$
3:     **if** $\|\mathbf{A}_k\mathbf{A}_k^\top - \bar{\boldsymbol{\Sigma}}\|_F^2 < \epsilon_1$ and $g(\mathbf{A}_k) < \epsilon_2$ **then break**
4:     Update penalty coefficient $c_{k+1} := \beta c_k$
5: **Output** solution $\mathbf{A}_k$

---

---

**Algorithm 2** Likelihood-Based Method

---

**Require:** initial penalty coefficient $c_1 > 0$; multiplicative factor $\beta > 1$; maximum number of iterations $k_{\max} > 0$; tolerance $\epsilon > 0$; initial solution $\mathbf{A}_0$; empirical covariance matrix $\bar{\boldsymbol{\Sigma}}$
1: **for** $k = 1, 2, \ldots, k_{\max}$ **do**
2:     Solve $\mathbf{A}_k := \arg\min_{\mathbf{A} \in \mathbb{R}^{n \times n}} L(\mathbf{A}; \bar{\boldsymbol{\Sigma}}) + \rho(\mathbf{A}) + \frac{c_k}{2}g(\mathbf{A})^2$ initialized at $\mathbf{A}_{k-1}$
3:     **if** $g(\mathbf{A}_k) < \epsilon$ **then break**
4:     Update penalty coefficient $c_{k+1} := \beta c_k$
5: **Output** solution $\mathbf{A}_k$

---

### 4.2 Estimation Methods

Based on the identifiabiltiy results of Theorems 1 and 6, we propose two estimation methods, called *SparseICA*, to perform ICA from second-order statistics that leverage sparsity regularization and continuous constrained optimization. To proceed, we define $\bar{\boldsymbol{\Sigma}}$ as the empirical covariance matrix of observed variables $\mathbf{x}$ and $T$ as the sample size.

**Decomposition-based method.** Given the formulation in Eq. (5), we consider the following constrained optimization problem

$$\min_{\mathbf{A} \in \mathbb{R}^{n \times n}} \rho(\mathbf{A}) \quad \text{subject to} \quad \mathbf{A}\mathbf{A}^\top - \bar{\boldsymbol{\Sigma}} = \mathbf{0} \quad \text{and} \quad g(\mathbf{A}) = 0, \tag{6}$$

where $\rho(\mathbf{A})$ is a suitable sparsity regularizer, often expressible as $\rho(\mathbf{A}) = \sum_{i,j} \rho(a_{i,j})$. Formulation (5) indicates that one should apply the $\ell_0$ regularizer $\rho(\mathbf{A}) = \|\mathbf{A}\|_0$. Alternatively, other possible choices include the $\ell_1$ regularizer $\rho(\mathbf{A}) = \|\mathbf{A}\|_1$ that supports continuous optimization. Further details regarding our specific choice of sparsity regularizer will be elaborated later in this section. On the other hand, we simply use the empirical covariance matrix $\bar{\boldsymbol{\Sigma}}$ as an estimate of the true covariance matrix $\tilde{\boldsymbol{\Sigma}}$, which is found to work well across different sample sizes in our experiments. One may also adopt a regularized estimator of the form $\bar{\boldsymbol{\Sigma}} + \eta \mathbf{I}$ with a proper choice of $\eta$, which may have notable advantage in certain cases [13, 28, 37].

**Likelihood-based method.** In addition to the decomposition-based method above, we introduce a likelihood-based estimation method formulated by the following constrained optimization problem:

$$\min_{\mathbf{A} \in \mathbb{R}^{n \times n}} L(\mathbf{A}; \bar{\boldsymbol{\Sigma}}) + \rho(\mathbf{A}) \quad \text{subject to} \quad g(\mathbf{A}) = 0, \tag{7}$$

$$\text{where} \quad L(\mathbf{A}; \bar{\boldsymbol{\Sigma}}) = \frac{T}{2}\operatorname{tr}((\mathbf{A}^\top\mathbf{A})^{-1}\bar{\boldsymbol{\Sigma}}) + T\log|\det\mathbf{A}|$$

is the negative Gaussian log-likelihood function and $\rho(\mathbf{A})$ is a sparsity regularizer. The following result establishes the theoretical guarantee of this likelihood-based method.

**Theorem 7 (Likelihood-Based Method).** *Suppose that the true mixing matrix $\tilde{\mathbf{A}}$ satisfies Assumptions 1, 2, and 3. Let $\hat{\mathbf{A}}$ be a solution of Problem (7) with sparsity regularizer $\rho(\mathbf{A}) = 0.5\|\mathbf{A}\|_0 \log T$. Then, we have $\hat{\mathbf{A}} \sim \tilde{\mathbf{A}}$ in the large sample limit.*

**Implementation.** Based on Theorems 6 and 7, ideally one should adopt the $\ell_0$ regularizer $\rho(\mathbf{A}) = \lambda\|\mathbf{A}\|_0$ and develop an exact discrete search procedure over the support space of matrix $\mathbf{A}$. However, such approach may pose computational challenges in practice. Since the functions $L(\mathbf{A}; \bar{\boldsymbol{\Sigma}})$ and $g(\mathbf{A})$ are differentiable, in this work we develop an estimation procedure that leverages efficient continuous optimization techniques. Therefore, some possible choices for $\rho(\mathbf{A})$ are $\ell_1$, smoothly

clipped absolute deviation (SCAD) [17], and minimax concave penalty (MCP) [46] regularizers. The $\ell_1$ regularizer has been shown to exhibit bias during estimation [17, 10], especially for large coefficients. Here, we adopt the MCP regularizer that is less susceptible to such issue, given by

$$\rho(a_{i,j}) = \begin{cases} \lambda|a_{i,j}| - \frac{a_{i,j}^2}{2\alpha}, & \text{if } |a_{i,j}| \leq \alpha\lambda, \\ \frac{\alpha\lambda^2}{2}, & \text{otherwise,} \end{cases}$$

where $\lambda$ and $\alpha$ are hyperparameters.

To solve Eqs. (6) and (7), standard constrained optimization methods can be used, such as quadratic penalty method, augmented Lagrangian method, and barrier method [7, 8, 33]. In this work, we adopt the quadratic penalty method that converts each constrained problem into a sequence of unconstrained optimization problems where the constraint violations are increasingly penalized. We describe the full procedure of the decomposition-based and likelihood-based methods based on quadratic penalty method in Algorithms 1 and 2, respectively. The unconstrained problem in each iteration can be solved using different continuous optimization solvers, including first-order methods such as gradient descent and steepest descent, as well as second-order methods such as quasi-Newton methods. In our experiments presented in the subsequent section, we employ L-BFGS [11], a quasi-Newton method, to solve the unconstrained optimization problem.

It is worth noting that the formulations in Eqs. (6) and (7) involve solving nonconvex optimization problems; in practice, the optimization procedure may return stationary points that correspond to suboptimal local solutions. Therefore, we run the method for a number of times and choose the final mixing matrix via model selection. Further details regarding the optimization procedure and implementation are provided in Appendix F.

## 5  Experiments

To empirically validate our proposed identifiability results, we carry out experiments under various settings. We also conduct ablation studies to verify the necessity of the proposed assumptions and include *FastICA* [23] as a representative baseline. Specifically, we consider the following methods:

- *SparseICA*: Decomposition-based (Eq. (6)) or likelihood-based (Eq. (7)) method on data where both Assumptions 1 and 2 hold;
- *Vanilla*: Decomposition-based (Eq. (6)) or likelihood-based (Eq. (7)) method without the constraint $g(\mathbf{A}) = 0$, on data where neither Assumption 1 nor Assumption 2 holds;
- *FastICA-D*: FastICA on data where both Assumptions 1 and 2 hold;
- *FastICA*: FastICA on data where neither Assumption 1 nor Assumption 2 holds.

For all experiments, we simulate 10 sources, and generate the supports of the true mixing matrices $\tilde{\mathbf{A}}$ according to the assumptions required by each method above. The nonzero entries of $\tilde{\mathbf{A}}$ are sampled uniformly at random from $[-0.8, -0.2] \cup [0.2, 0.8]$. We use mean correlation coefficient (MCC) and Amari distance [3] as evaluation metrics, where all results are reported for 10 random trials.

**Different sample sizes.**   We first consider entirely Gaussian sources and different sample sizes. The empirical results of MCC are shown in Figure 1, while those of Amari distance are given in Figure 4 in Appendix G. By comparing *SparseICA* with *Vanilla* and *FastICA*, it is evident that the identification performance is much better across different sample sizes when the required assumptions on the connective structure are satisfied, as validated by Wilcoxon signed-rank test at $5\%$ significance level. Furthermore, the unsatisfactory results of *FastICA-D* indicate that our estimation methods are also essential for ensuring the quality of the identification, which further validates the proposed identifiability theory. Since *FastICA-D* performs similarly to *FastICA*, it suggests that the data-generating process, while meeting our assumption, may not be inherently simpler to recover without considering specific procedure to handle Gaussian sources. In addition, as expected, the performance of SparseICA improves in terms of both MCC and Amari distance as the sample size increases.

**Different ratios of Gaussian sources.**   We now conduct empirical study to investigate the performance in the presence of Gaussian and non-Gaussian sources. Here, the non-Gaussian sources follow exponential distributions. We consider different ratios of Gaussian sources, which are specifically $0, 0.2, 0.4, 0.6, 0.8$, and $1$. For instance, ratio of $0.6$ indicates that there are 6 Gaussian sources

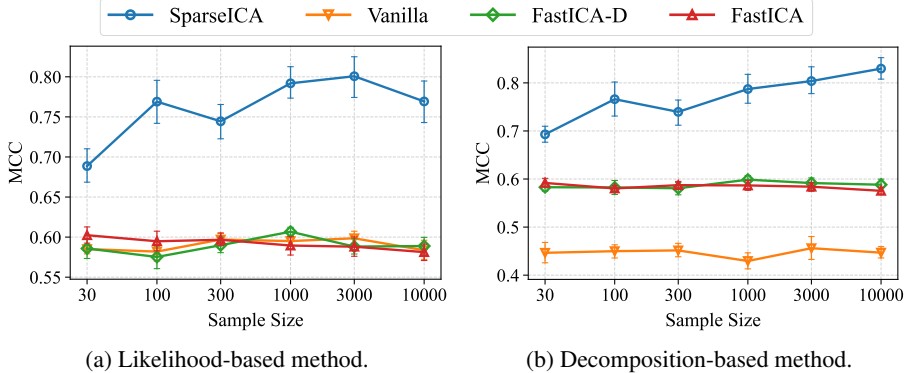

(a) Likelihood-based method.

(b) Decomposition-based method.

Figure 1: Empirical results of MCC across different sample sizes. Error bars indicate the standard errors calculated based on 10 random trials.

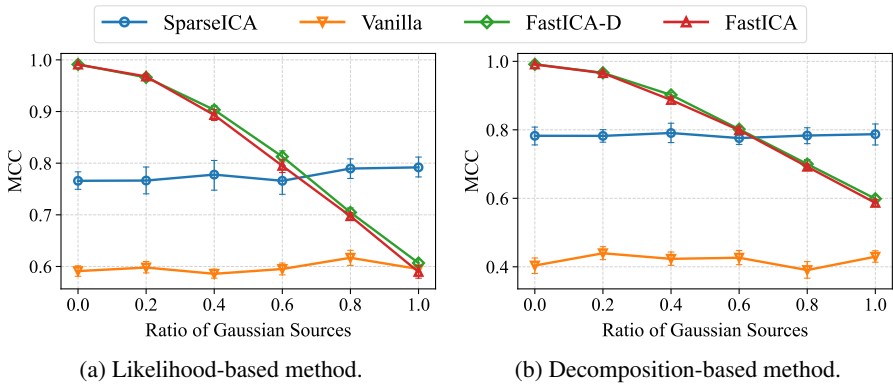

(a) Likelihood-based method.

(b) Decomposition-based method.

Figure 2: Empirical results of MCC across different ratios of Gaussian sources. Error bars indicate the standard errors calculated based on 10 random trials.

and 4 non-Gaussian sources. The empirical results of MCC based on 1000 samples are depicted in Figure 1, while those of Amari distance are provided in Figure 4 in Appendix G. One observes that the identification performance *SparseICA* is rather stable across different ratios of Gaussian sources, which may not be surprising as it leverages only second-order statistics. On the other hand, the performance of *FastICA-D* and *FastICA* deteriorates as the ratio of Gaussian sources increases, because it relies on non-Gaussianity of the sources. It is also observed that, in the presence of Gaussian sources, *FastICA-D* and *FastICA* may perform well, provided that the ratio (or number) of Gaussian sources is not large. This suggests a potential future direction to integrate our method based on second-order statistics with existing methods that rely on non-Gaussianity, which may better handle both Gaussian and non-Gaussian sources.

## 6   Conclusion

We develop an identifiability theory of ICA from second-order statistics without relying on non-Gaussianity. Specifically, we introduce novel and precise assumptions on the connective structure from sources and observed variables, and show that our proposed assumption of structural variability is strictly weaker than the previous ones. Importantly, we prove that this assumption is one of the necessary conditions for achieving identifiability in the investigated setting. We further propose two estimation methods based on second-order statistics that leverage sparsity regularization. Moreover, we establish a precise connection between our identifiability result of ICA and causal discovery from second-order statistics, which may open up avenues for exploring the interplay between ICA and causal discovery with linear Gaussian SEM. Our theoretical claims have also been empirically validated across different settings. The limitations include the lack of finite sample analysis and broader application of our theory in more real-world tasks, which are worth exploring in future work.

## Acknowledgments

The authors would like to thank the anonymous reviewers for helpful comments and suggestions. This project is partially supported by NSF Grant 2229881, the National Institutes of Health (NIH) under Contract R01HL159805, a grant from Apple Inc., a grant from KDDI Research Inc., and generous gifts from Salesforce Inc., Microsoft Research, and Amazon Research.

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

# Appendices

## Table of Contents

## A  Preliminaries on Support Rotation

Before presenting the proofs of the theoretical results, we review the definition of support rotation and its potential effects on a support matrix, given by Ghassami et al. [20]. Note that the definition and remark below are quoted from Ghassami et al. [20, Section 3.1], with only minor modifications.

**Definition 3 (Support Rotation [20]).** *The support rotation, denoted as $R(i, j, k)$, is a transformation that modifies a support matrix $\xi$ by applying a Givens rotation in the $(j, k)$ plane, setting the element $\xi_{i,j}$ to zero. The outcome of is the support matrix of $\mathbf{Q}G\left(j, k, \tan^{-1}\left(-q_{i,j}/q_{i,k}\right)\right)$, where $\mathbf{Q} \in \arg\max_{\mathbf{Q}'} \left|\text{supp}\left(\mathbf{Q}'G\left(j, k, \tan^{-1}\left(-q'_{i,j}/q'_{i,k}\right)\right)\right)\right|$ such that the support matrix of $\mathbf{Q}'$ is $\xi$.*

Note that $G\left(j, k, \tan^{-1}\left(-q'_{ij}/q'_{i,k}\right)\right)$ is the Givens rotation in the $(j, k)$ plane that makes the $q'_{i,j}$ entry zero.

**Remark 3 (Ghassami et al. [20]).** *The effects of applying a support rotation $R(i, j, k)$ can be categorized into the following four cases:*

- **Reduction:** *If $\xi_{i,j} = \xi_{i,k} = \times$ and $\xi_{l,j} = \xi_{l,k}$ for all $l \in [p]\backslash\{i\}$, then only $\xi_{i,j}$ becomes zero.*

- **Reversible acute rotation:** *If $\xi_{i,j} = \xi_{i,k} = \times$ and there exists a row $i'$ such that the $j$-th and $k$-th columns differ only in that row, then $\xi_{i,j}$ becomes zero and both $\xi_{i',j}$ and $\xi_{i',k}$ become $\times$.*

- **Irreversible acute rotation:** *If $\xi_{i,j} = \xi_{i,k} = \times$ and the $j$-th and $k$-th columns differ in at least two rows, then $\xi_{i,j}$ becomes zero and all entries on the $j$-th and $k$-th columns become $\times$ on the rows on which they differed.*

- **Column swap:** *If $\xi_{i,j} = \times$ and $\xi_{i,k} = 0$, then columns $j$ and $k$ are swapped.*

# B   Proofs of Useful Lemmas

We provide several lemmas that will be useful for subsequent proofs.

## B.1   Proof of Lemma 2

**Lemma 2 (Nonzero Diagonal Entries).**  *For any non-singular matrix $\mathbf{A}$, there exists a permutation matrix $\mathbf{P}$ such that the diagonal entries of $\mathbf{AP}$ are nonzero.*

*Proof.*  We prove it by contradiction. Suppose that there always exists a zero diagonal entry in every column permutation.

We first represent the determinant of the matrix $\mathbf{A}$ as its Leibniz formula:

$$\det(\mathbf{A}) = \sum_{\sigma \in \mathcal{S}_n} \left( \text{sgn}(\sigma) \prod_{i=1}^{n} a_{i,\sigma(i)} \right),$$

where $\mathcal{S}_n$ is the set of $n$-permutations. Because for every permutation, there always exists a diagonal entry with value zero, we have

$$\prod_{i=1}^{n} a_{i,\sigma(i)} = 0, \quad \forall \sigma \in \mathcal{S}_n.$$

Thus, it follows that $\det(\mathbf{A}) = 0$. This indicates that the matrix $\mathbf{A}$ is singular, which leads to a contradiction. □

## B.2   Proof of Lemma 3

**Lemma 3 (Non-Singular Solution).**  *Let $\tilde{\mathbf{A}}$ be a non-singular matrix. Suppose $\hat{\mathbf{A}}$ is a matrix such that $\hat{\mathbf{A}}\hat{\mathbf{A}}^{\top} = \tilde{\mathbf{A}}\tilde{\mathbf{A}}^{\top}$. Then, $\hat{\mathbf{A}}$ is non-singular.*

*Proof.*  A matrix is non-singular (or invertible) if and only if its determinant is nonzero. Therefore, we need to show that $\det(\hat{\mathbf{A}}) \neq 0$.

From the given, we have $\hat{\mathbf{A}}\hat{\mathbf{A}}^{\top} = \tilde{\mathbf{A}}\tilde{\mathbf{A}}^{\top}$. Taking determinants on both sides, we get

$$\det(\hat{\mathbf{A}}\hat{\mathbf{A}}^{\top}) = \det(\tilde{\mathbf{A}}\tilde{\mathbf{A}}^{\top}).$$

Since $\hat{\mathbf{A}}$ and $\tilde{\mathbf{A}}$ are square matrices, we can simplify both sides:

$$\det(\hat{\mathbf{A}})^2 = \det(\tilde{\mathbf{A}})^2.$$

Since $\tilde{\mathbf{A}}$ is non-singular, we know that $\det(\tilde{\mathbf{A}}) \neq 0$. Therefore, $\det(\hat{\mathbf{A}})^2 \neq 0$. This implies that $\det(\hat{\mathbf{A}}) \neq 0$. Thus, $\hat{\mathbf{A}}$ is non-singular. □

## B.3 Proof of Lemma 5

**Lemma 4.** *Let* $\mathbf{A}$ *be a matrix with all diagonal entries being nonzero. Then, there exists matrix* $\mathbf{B} \in \mathbb{R}_{\text{off}}^{n \times n}$ *and* $\boldsymbol{\Omega} \in \text{diag}(\mathbb{R}_{>0}^n)$ *such that* $\mathbf{A} = (\mathbf{I} - \mathbf{B})\boldsymbol{\Omega}^{-\frac{1}{2}}\mathbf{D}$, *where* $\mathbf{D}$ *is a diagonal matrix with diagonal entries being* $\pm 1$.

*Proof.* Since the diagonal entries of $\mathbf{A}$ are nonzero, we can construct a diagonal matrix $\mathbf{D}$ such that the diagonal entries of $\mathbf{AD}$ are positive, by defining $d_{i,i} = 1$ if $a_{i,i} > 0$ and $d_{i,i} = -1$ if $a_{i,i} < 0$. Let $\boldsymbol{\Omega}$ be a diagonal matrix of the same size as matrix $\mathbf{A}$, where $(\boldsymbol{\Omega})_{i,i} = 1/(\mathbf{AD})_{i,i}^2$, Also, let $\mathbf{B} := \mathbf{I} - \mathbf{AD}\boldsymbol{\Omega}^{\frac{1}{2}}$. Since the diagonal entries of matrix $\mathbf{AD}\boldsymbol{\Omega}^{\frac{1}{2}}$ are ones, the diagonal entries of matrix $\mathbf{B}$ are zeros. Therefore, we have $\mathbf{B} \in \mathbb{R}_{\text{off}}^{n \times n}$, $\boldsymbol{\Omega} \in \text{diag}(\mathbb{R}_{>0}^n)$, and $\mathbf{A} = (\mathbf{I} - \mathbf{B})\boldsymbol{\Omega}^{-\frac{1}{2}}\mathbf{D}$, where $\mathbf{D}$ is a diagonal matrix with entries being $\pm 1$. $\qquad\square$

**Lemma 5.** *Let* $\mathbf{A}$ *be a non-singular matrix. Then, there exists matrix* $\mathbf{B} \in \mathbb{R}_{\text{off}}^{n \times n}$ *and* $\boldsymbol{\Omega} \in \text{diag}(\mathbb{R}_{>0}^n)$ *such that* $\mathbf{A} \sim (\mathbf{I} - \mathbf{B})\boldsymbol{\Omega}^{-\frac{1}{2}}$.

*Proof.* Since matrix $\mathbf{A}$ is non-singular, by Lemma 2, there exists a permutation matrix $\mathbf{P}$ such that the diagonal entries of $\mathbf{AP}$ are nonzero. By Lemma 4, there exists matrix $\mathbf{B} \in \mathbb{R}_{\text{off}}^{n \times n}$ and $\boldsymbol{\Omega} \in \text{diag}(\mathbb{R}_{>0}^n)$ such that $\mathbf{AP} = (\mathbf{I} - \mathbf{B})\boldsymbol{\Omega}^{-\frac{1}{2}}\mathbf{D}$, where $\mathbf{D}$ is a diagonal matrix with diagonal entries being $\pm 1$. Clearly, we have $\mathbf{A} \sim (\mathbf{I} - \mathbf{B})\boldsymbol{\Omega}^{-\frac{1}{2}}$. $\qquad\square$

## B.4 Proof of Lemma 6

**Lemma 6.** *Suppose* $\mathbf{A} \sim (\mathbf{I} - \mathbf{B})\boldsymbol{\Omega}^{-\frac{1}{2}}$ *for matrices* $\mathbf{A} \in \mathbb{R}^{n \times n}$, $\mathbf{B} \in \mathbb{R}_{\text{off}}^{n \times n}$, *and* $\boldsymbol{\Omega} \in \text{diag}(\mathbb{R}_{>0}^n)$. *Then, we have* $\|\mathbf{A}\|_0 = \|\mathbf{B}\|_0 + n$.

*Proof.* First notice that the diagonal entries of $\mathbf{B}$ are zeros, which implies

$$\|\mathbf{I} - \mathbf{B}\|_0 = \|\mathbf{I}\|_0 + \|\mathbf{B}\|_0 = \|\mathbf{B}\|_0 + n.$$

Since $\boldsymbol{\Omega}^{-\frac{1}{2}}$ is a diagonal matrix, right multiplication of $\boldsymbol{\Omega}^{-\frac{1}{2}}$ amounts to rescaling the columns of $\mathbf{I} - \mathbf{B}$, and does not affect its support. Therefore, we have

$$\|(\mathbf{I} - \mathbf{B})\boldsymbol{\Omega}^{-\frac{1}{2}}\|_0 = \|\mathbf{I} - \mathbf{B}\|_0 = \|\mathbf{B}\|_0 + n.$$

Since $\mathbf{A} \sim (\mathbf{I} - \mathbf{B})\boldsymbol{\Omega}^{-\frac{1}{2}}$, matrices $\mathbf{A}$ and $(\mathbf{I} - \mathbf{B})\boldsymbol{\Omega}^{-\frac{1}{2}}$ differ only in signed column permutations. Therefore, their number of nonzero entries are the same, i.e.,

$$\|\mathbf{A}\|_0 = \|\mathbf{B}\|_0 + n. \qquad\square$$

## B.5 Proof of Lemma 8

We first provide the following lemma that will be used to prove Lemma 8.

**Lemma 7.** *Let* $\mathbf{P}_1$ *and* $\mathbf{P}_2$ *be two permutation matrices. Then,* $\mathbf{P}_1^\top \mathbf{P}_2$ *is lower triangular if and only if* $\mathbf{P}_1 = \mathbf{P}_2$.

*Proof.* The "if part" is clear because permutation matrices are orthogonal matrices, and thus $\mathbf{P}_1^\top \mathbf{P}_1 = \mathbf{I}$ is lower triangular. It remains to prove the "only if part". Since $\mathbf{P}_1^\top \mathbf{P}_2$ is also a permutation matrix, this matrix being lower triangular implies that it is an identity matrix, because identity matrix is the only permutation matrix that is lower triangular. We then have $\mathbf{P}_1^\top \mathbf{P}_2 = \mathbf{I}$, which implies

$$\mathbf{P}_1 = \mathbf{P}_2^{-\top} = \mathbf{P}_2. \qquad\square$$

We now provide the proof of Lemma 8.

**Lemma 8.** *Given matrix* $\mathbf{B} \in \mathbb{R}_{\text{off}}^{n \times n}$, *if* $\mathbf{I} - \mathbf{B}$ *satisfies Assumption 2, then* $\mathbf{I} - \mathbf{B}$ *is non-singular.*

*Proof.* Since $\mathbf{I} - \mathbf{B}$ satisfies Assumption 2, there exist permutation matrices $\mathbf{P}_1$ and $\mathbf{P}_2$ such that

$$\mathbf{P}_1^\top(\mathbf{I} - \mathbf{B})\mathbf{P}_2 = \mathbf{P}_1^\top \mathbf{P}_2 - \mathbf{P}_1^\top \mathbf{B}\mathbf{P}_2$$

is lower triangular. For all $i, j \in [n]$ such that $(\mathbf{P}_1^\top \mathbf{P}_2)_{i,j} \neq 0$, we have $(\mathbf{P}_1^\top \mathbf{B}\mathbf{P}_2)_{i,j} = 0$, which implies that $\mathbf{P}_1^\top \mathbf{P}_2$ must be lower triangular, because otherwise, $\mathbf{P}_1^\top \mathbf{P}_2 - \mathbf{P}_1^\top \mathbf{B}\mathbf{P}_2$ cannot be lower

triangular. By Lemma 7, we then have $\mathbf{P}_1 = \mathbf{P}_2$. This indicates that $\mathbf{I} - \mathbf{P}_1^\top \mathbf{B} \mathbf{P}_1$ is lower triangular, with diagonal entries equal to one. Therefore, we have

$$\det(\mathbf{I} - \mathbf{P}_1^\top \mathbf{B} \mathbf{P}_1) = 1,$$

which implies

$$\det(\mathbf{I} - \mathbf{B}) = \det(\mathbf{P}_1(\mathbf{I} - \mathbf{B})\mathbf{P}_1^\top) = \det(\mathbf{I} - \mathbf{P}_1^\top \mathbf{B} \mathbf{P}_1) = 1.$$

Since the determinant of $\mathbf{I} - \mathbf{B}$ is nonzero, it is non-singular. $\square$

## C  Proof of Theorem 1

We first describe the notion of covariance equivalence that is needed for the proof of Theorem 1. Specifically, if two support matrices $\boldsymbol{\xi}_1$ and $\boldsymbol{\xi}_2$ entail the same set of covariance matrices, i.e., $\boldsymbol{\Sigma}(\boldsymbol{\xi}_1) = \boldsymbol{\Sigma}(\boldsymbol{\xi}_2)$, they are said to be *covariance equivalent*. This means that for any combination of parameter values in $\boldsymbol{\xi}_1$, there exists a corresponding set of parameter values in $\boldsymbol{\xi}_2$ that leads to the same covariance matrix, and vice versa. Furthermore, two support matrices are covariance equivalent if and only if they lead to the same set of semialgebraic constraints on the covariance matrices.

We now give the proof of Theorem 1. The proof makes use of Proposition 2, Proposition 6, Proposition 7, and Corollary 1, which are provided in Appendices C.1, C.2, C.3, and C.4, respectively. Note that the proof is partly inspired by that of Ghassami et al. [20, Theorem 3].

**Theorem 1 (Identifiability with Sparsity).** *Suppose that the true mixing matrix $\tilde{\mathbf{A}}$ satisfies Assumptions 1, 2, and 3. Let $\hat{\mathbf{A}}$ be a solution of the following problem:*

$$\min_{\mathbf{A} \in \mathbb{R}^{n \times n}} \|\mathbf{A}\|_0 \quad \text{subject to} \quad \mathbf{A}\mathbf{A}^\top = \tilde{\mathbf{A}}\tilde{\mathbf{A}}^\top \quad \text{and} \quad \mathbf{A} \text{ satisfies Assumption 2.} \tag{2}$$

*Then, we have $\hat{\mathbf{A}} \sim \tilde{\mathbf{A}}$.*

*Proof.* Let $\hat{\mathbf{A}}$ be a solution of Problem (2). This implies $\hat{\mathbf{A}}\hat{\mathbf{A}}^\top = \tilde{\boldsymbol{\Sigma}} = \tilde{\mathbf{A}}\tilde{\mathbf{A}}^\top$ and that $\hat{\mathbf{A}}$ satisfies Assumption 2. Since $\tilde{\mathbf{A}}$ is non-singular, by Lemma 3, matrix $\hat{\mathbf{A}}$ is non-singular.

Since $\hat{\mathbf{A}}$ can entail the covariance matrix $\tilde{\boldsymbol{\Sigma}}$, we have $\tilde{\boldsymbol{\Sigma}} \in \boldsymbol{\Sigma}(\boldsymbol{\xi}_{\hat{\mathbf{A}}})$, which indicates that $\tilde{\boldsymbol{\Sigma}}$ contains all semialgebraic constraints of $\boldsymbol{\xi}_{\hat{\mathbf{A}}}$. Under Assumption 3, we have

$$H(\boldsymbol{\xi}_{\hat{\mathbf{A}}}) \subseteq H(\boldsymbol{\xi}_{\tilde{\mathbf{A}}}). \tag{8}$$

The sparsity term in the objective function implies

$$\|\hat{\mathbf{A}}\|_0 \leq \|\tilde{\mathbf{A}}\|_0, \tag{9}$$

because otherwise $\hat{\mathbf{A}}$ will never be a solution of Problem (2).

We now show by contradiction that $H(\boldsymbol{\xi}_{\hat{\mathbf{A}}}) \not\subset H(\boldsymbol{\xi}_{\tilde{\mathbf{A}}})$. Suppose $H(\boldsymbol{\xi}_{\hat{\mathbf{A}}}) \subset H(\boldsymbol{\xi}_{\tilde{\mathbf{A}}})$, which indicates $\dim(\boldsymbol{\Sigma}(\boldsymbol{\xi}_{\hat{\mathbf{A}}})) > \dim(\boldsymbol{\Sigma}(\boldsymbol{\xi}_{\tilde{\mathbf{A}}}))$. Since the support matrices $\boldsymbol{\xi}_{\hat{\mathbf{A}}}$ and $\boldsymbol{\xi}_{\tilde{\mathbf{A}}}$ satisfy Assumption 2, Proposition 2 implies $\|\boldsymbol{\xi}_{\hat{\mathbf{A}}}\|_0 > \|\boldsymbol{\xi}_{\tilde{\mathbf{A}}}\|_0$, which is contradictory with Inequality (9). This implies

$$H(\boldsymbol{\xi}_{\hat{\mathbf{A}}}) \not\subset H(\boldsymbol{\xi}_{\tilde{\mathbf{A}}}). \tag{10}$$

By Eqs. (8) and (10), we obtain $H(\boldsymbol{\xi}_{\hat{\mathbf{A}}}) = H(\boldsymbol{\xi}_{\tilde{\mathbf{A}}})$, which, by Proposition 7, implies that the support matrices $\boldsymbol{\xi}_{\hat{\mathbf{A}}}$ and $\boldsymbol{\xi}_{\tilde{\mathbf{A}}}$ are covariance equivalent, because they satisfy Assumption 2. By Proposition 6, we conclude that the columns of $\boldsymbol{\xi}_{\hat{\mathbf{A}}}$ are a permutation of those of $\boldsymbol{\xi}_{\tilde{\mathbf{A}}}$.

Recall that matrices $\tilde{\mathbf{A}}$ and $\hat{\mathbf{A}}$ are non-singular, and entail the same covariance matrix $\tilde{\boldsymbol{\Sigma}}$. Since the columns of $\boldsymbol{\xi}_{\hat{\mathbf{A}}}$ are a permutation of those of $\boldsymbol{\xi}_{\tilde{\mathbf{A}}}$, by Corollary 1, we have $\hat{\mathbf{A}} \sim \tilde{\mathbf{A}}$. $\square$

### C.1  Dimension of Covariance Set

We first define the Jacobian matrix of $\boldsymbol{\Sigma} = \mathbf{A}\mathbf{A}^\top$ w.r.t. the free parameters of $\mathbf{A}$ as $\frac{\partial \boldsymbol{\Sigma}}{\partial \mathbf{A}}$, which is a $n^2 \times \|\mathbf{A}\|_0$ matrix, where the rows are indexed by $(i, j) \in [n] \times [n]$, and columns are indexed by

$(k,l) \in \mathrm{supp}(\mathbf{A})$. That is, for $(i,j) \in [n] \times [n]$ and $(k,l) \in \mathrm{supp}(\mathbf{A})$, we have

$$\left(\frac{\partial \boldsymbol{\Sigma}}{\partial \mathbf{A}}\right)_{(i,j),(k,l)} = \frac{\partial \Sigma_{i,j}}{\partial a_{k,l}} = \begin{cases} 2a_{i,l} & \text{if } i = j \text{ and } k = i, \\ 0 & \text{if } i = j \text{ and } k \neq i, \\ a_{j,l} & \text{if } i \neq j \text{ and } k = i, \\ a_{i,l} & \text{if } i \neq j \text{ and } k = j, \\ 0 & \text{if } i \neq j \text{ and } k \notin \{i,j\}. \end{cases} \tag{11}$$

Given a matrix $\mathbf{M}$, we denote by $\mathrm{off}(\mathbf{M})$ and $\mathrm{on}(\mathbf{M})$ the off-diagonal and diagonal entries of $\mathbf{M}$, respectively. With a slight abuse of notation, we rewrite the above Jacobian matrix as the following form that consists of four submatrices, by permuting the corresponding columns and rows:

$$\frac{\partial \boldsymbol{\Sigma}}{\partial \mathbf{A}} = \begin{bmatrix} \frac{\partial \,\mathrm{off}(\boldsymbol{\Sigma})}{\partial \,\mathrm{off}(\mathbf{A})} & \frac{\partial \,\mathrm{off}(\boldsymbol{\Sigma})}{\partial \,\mathrm{on}(\mathbf{A})} \\ \frac{\partial \,\mathrm{on}(\boldsymbol{\Sigma})}{\partial \,\mathrm{off}(\mathbf{A})} & \frac{\partial \,\mathrm{on}(\boldsymbol{\Sigma})}{\partial \,\mathrm{on}(\mathbf{A})} \end{bmatrix}. \tag{12}$$

Each submatrix above represents the Jacobian matrix for different parts (i.e., off-diagonal and diagonal entries) of matrix $\boldsymbol{\Sigma}$ taken w.r.t. different free parameters (i.e., off-diagonal and diagonal free parameters) in matrix $\mathbf{A}$. Note that column and row permutations do not affect the rank of the matrix. Since we are primarily interested in the rank of the above Jacobian matrix and its submatrices, we use the same notation to refer to different column and/or row permutations of the corresponding Jacobian matrix, depending on the context.

**Lemma 9.** *Let $\mathbf{A}$ be a lower triangular matrix. Then, we have*

$$\mathrm{rank}\left(\left.\frac{\partial \boldsymbol{\Sigma}}{\partial \mathbf{A}}\right|_{\mathbf{A}=\mathbf{I}_n}\right) = \|\mathbf{A}\|_0.$$

*Proof.* Let $d$ be the number of diagonal free parameters in matrix $\mathbf{A}$. This indicates that $\mathrm{off}(\mathbf{A})$ and $\mathrm{on}(\mathbf{A})$ contain $\|\mathbf{A}\|_0 - d$ and $d$ free parameters, respectively.

By specializing $\mathbf{A} = \mathbf{I}_n$ and using Eq. (11), we have

$$\left.\frac{\partial \,\mathrm{off}(\boldsymbol{\Sigma})}{\partial \,\mathrm{on}(\mathbf{A})}\right|_{\mathbf{A}=\mathbf{I}_n} = \mathbf{0}, \quad \left.\frac{\partial \,\mathrm{on}(\boldsymbol{\Sigma})}{\partial \,\mathrm{off}(\mathbf{A})}\right|_{\mathbf{A}=\mathbf{I}_n} = \mathbf{0}, \quad \text{and} \quad \left.\frac{\partial \,\mathrm{on}(\boldsymbol{\Sigma})}{\partial \,\mathrm{on}(\mathbf{A})}\right|_{\mathbf{A}=\mathbf{I}_n} = 2\mathbf{I}_d, \tag{13}$$

where the last equality is obtained after row permutations of the corresponding matrix. Also, since matrix $\mathbf{A}$ is lower triangular, each nonzero entry of $\left.\frac{\partial \,\mathrm{off}(\boldsymbol{\Sigma})}{\partial \,\mathrm{off}(\mathbf{A})}\right|_{\mathbf{A}=\mathbf{I}_n}$ corresponds to either

$$\frac{\partial \Sigma_{k,l}}{\partial a_{k,l}} = 1 \quad \text{or} \quad \frac{\partial \Sigma_{l,k}}{\partial a_{k,l}} = 1 \quad \text{for some } k > l.$$

In this case, each column of $\left.\frac{\partial \,\mathrm{off}(\boldsymbol{\Sigma})}{\partial \,\mathrm{off}(\mathbf{A})}\right|_{\mathbf{A}=\mathbf{I}_n}$ contains precisely two nonzero entries, while each row either contains precisely one nonzero entry, or does not contain any nonzero entry. Therefore, $\left.\frac{\partial \,\mathrm{off}(\boldsymbol{\Sigma})}{\partial \,\mathrm{off}(\mathbf{A})}\right|_{\mathbf{A}=\mathbf{I}_n}$ can be rewritten after row permutations as

$$\left.\frac{\partial \,\mathrm{off}(\boldsymbol{\Sigma})}{\partial \,\mathrm{off}(\mathbf{A})}\right|_{\mathbf{A}=\mathbf{I}_n} = \begin{bmatrix} \mathbf{I}_{\|\mathbf{A}\|_0 - d} \\ \mathbf{I}_{\|\mathbf{A}\|_0 - d} \\ \mathbf{0} \end{bmatrix},$$

which yields

$$\mathrm{rank}\left(\left.\frac{\partial \,\mathrm{off}(\boldsymbol{\Sigma})}{\partial \,\mathrm{off}(\mathbf{A})}\right|_{\mathbf{A}=\mathbf{I}_n}\right) = \|\mathbf{A}\|_0 - d. \tag{14}$$

Substituting Eq. (13) into Eq. (12), we have

$$\left.\frac{\partial \boldsymbol{\Sigma}}{\partial \mathbf{A}}\right|_{\mathbf{A}=\mathbf{I}_n} = \begin{bmatrix} \left.\frac{\partial \,\mathrm{off}(\boldsymbol{\Sigma})}{\partial \,\mathrm{off}(\mathbf{A})}\right|_{\mathbf{A}=\mathbf{I}_n} & \mathbf{0} \\ \mathbf{0} & 2\mathbf{I}_d \end{bmatrix},$$

which, with Eq. (14), implies

$$\begin{aligned} \mathrm{rank}\left(\left.\frac{\partial \boldsymbol{\Sigma}}{\partial \mathbf{A}}\right|_{\mathbf{A}=\mathbf{I}_n}\right) &= \mathrm{rank}\left(\left.\frac{\partial \,\mathrm{off}(\boldsymbol{\Sigma})}{\partial \,\mathrm{off}(\mathbf{A})}\right|_{\mathbf{A}=\mathbf{I}_n}\right) + \mathrm{rank}(2\mathbf{I}_d) \\ &= \|\mathbf{A}\|_0 - d + d \\ &= \|\mathbf{A}\|_0. \end{aligned}$$

$\square$

**Proposition 2 (Dimension of Covariance Set).** *Let $\boldsymbol{\xi}$ be a support matrix that satisfies Assumption 2. Then, its covariance set has a dimension of $\|\boldsymbol{\xi}\|_0$, i.e., $\dim(\boldsymbol{\Sigma}(\boldsymbol{\xi})) = \|\boldsymbol{\xi}\|_0$.*

*Proof.* Since $\boldsymbol{\xi}$ satisfies Assumption 2, it can be permuted by column and row permutations to be lower triangular; in this case, the resulting covariance matrix and the original covariance matrix differ in equal row and column permutations. Note that the dimension of the covariance set remains the same after simultaneous equal row and column permutations of the covariance matrices. Therefore, it suffices to consider the case where $\boldsymbol{\xi}$ is lower triangular, and show that it has a dimension of $\|\boldsymbol{\xi}\|_0$.

As indicated by Geiger et al. [19, Theorem 10], the dimension of $\dim(\boldsymbol{\Sigma}(\boldsymbol{\xi}))$ equals the maximum rank of the corresponding Jacobian matrix. In this case, it suffices to consider the columns of the Jacobian matrix that correspond to the nonzero entries of support matrix $\boldsymbol{\xi}$, i.e., the free parameters of matrix $\mathbf{A}$, which we denote by $\frac{\partial \boldsymbol{\Sigma}}{\partial \mathbf{A}}$. By Lemma 9, when $\mathbf{A} = \mathbf{I}_n$, the Jacobian matrix $\frac{\partial \boldsymbol{\Sigma}}{\partial \mathbf{A}}$ has full column rank that is equal to $\|\boldsymbol{\xi}\|_0$. Therefore, the dimension of the covariance set is $\|\boldsymbol{\xi}\|_0$. $\qquad\square$

## C.2 Covariance Equivalence and Column Permutation of Support

We now state a result that is adapted from Ghassami et al. [20, Proposition 5] to the context of ICA. In our proof of Theorem 1, only the "only if part" of the following result is used.

**Proposition 6 (Ghassami et al. [20, Proposition 5]).** *Consider two support matrices $\boldsymbol{\xi}_1$ and $\boldsymbol{\xi}_2$. If every pair of columns of $\boldsymbol{\xi}_1$ differ in more than one entry, then $\boldsymbol{\xi}_1$ and $\boldsymbol{\xi}_2$ are covariance equivalent if and only if the columns of $\boldsymbol{\xi}_2$ are a permutation of columns of $\boldsymbol{\xi}_1$.*

## C.3 Equality Constraints and Covariance Equivalence

In this section, we show how Assumption 2 allows one to go from two support matrices $\boldsymbol{\xi}_1$ and $\boldsymbol{\xi}_2$ having the same equality constraints to covariance equivalence.

Following Ghassami et al. [20], we first denote the covariance set of a directed graph $\mathcal{G}$ by

$$\boldsymbol{\Theta}(\mathcal{G}) := \{(\mathbf{I} - \mathbf{B})\boldsymbol{\Omega}^{-1}(\mathbf{I} - \mathbf{B})^\top : \mathbf{B} \in \mathbb{R}^{n \times n}_{\text{off}}, \boldsymbol{\Omega} \in \text{diag}(\mathbb{R}^n_{>0}), \text{supp}(\mathbf{B}) \subseteq \text{supp}(\mathbf{B}_\mathcal{G})\},$$

where $\mathbf{B}_\mathcal{G}$ is the adjacency matrix of $\mathcal{G}$. With a slight abuse of notation, we denote by $H(\mathcal{G})$ the set of the equality constraints imposed by $\mathcal{G}$ on the resulting matrix $(\mathbf{I} - \mathbf{B})\boldsymbol{\Omega}^{-1}(\mathbf{I} - \mathbf{B})^\top$.

**Lemma 10.** *Let $\boldsymbol{\xi}$ be a non-singular support matrix that satisfies Assumption 2.[2] Then, there exists a DAG $\mathcal{G}$ such that $\boldsymbol{\Sigma}(\boldsymbol{\xi}) = \boldsymbol{\Theta}(\mathcal{G})$.*

*Proof.* Since $\boldsymbol{\xi}$ is non-singular, by Lemma 2, it can be mapped via column permutations to another support matrix $\boldsymbol{\xi}'$ with diagonal entries being nonzero. Since $\boldsymbol{\xi}$ satisfies Assumption 2, Proposition 10 implies that $\boldsymbol{\xi}'$ represents a DAG, say $\mathcal{G}$, where the support of its adjacency matrix is $\text{supp}(\mathbf{B}_\mathcal{G}) = \text{supp}(\text{off}(\boldsymbol{\xi}'))$. Since column permutations of support matrices do not affect the resulting covariance matrices, we have $\boldsymbol{\Sigma}(\boldsymbol{\xi}) = \boldsymbol{\Sigma}(\boldsymbol{\xi}')$. Therefore, it suffices to prove $\boldsymbol{\Sigma}(\boldsymbol{\xi}') = \boldsymbol{\Theta}(\mathcal{G})$.

Since $\boldsymbol{\xi}'$ is a support matrix with diagonal entries being nonzero, we have

$$\text{supp}(\boldsymbol{\xi}') = \text{supp}(\mathbf{I}) \cup \text{supp}(\text{off}(\boldsymbol{\xi}')). \tag{15}$$

We consider both parts of the statements.

**Proof of $\boldsymbol{\Theta}(\mathcal{G}) \subseteq \boldsymbol{\Sigma}^+(\boldsymbol{\xi}')$:**

Suppose $\mathbf{M} \in \boldsymbol{\Theta}(\mathcal{G})$. By definition of $\boldsymbol{\Theta}(\mathcal{G})$, there exist $\mathbf{B} \in \mathbb{R}^{n \times n}_{\text{off}}$ and $\boldsymbol{\Omega} \in \text{diag}(\mathbb{R}^n_{>0})$ such that

$$\text{supp}(\mathbf{B}) \subseteq \text{supp}(\text{off}(\boldsymbol{\xi}')) \quad \text{and} \quad \mathbf{M} = (\mathbf{I} - \mathbf{B})\boldsymbol{\Omega}^{-1}(\mathbf{I} - \mathbf{B})^\top. \tag{16}$$

Let $\mathbf{A} := (\mathbf{I} - \mathbf{B})\boldsymbol{\Omega}^{-\frac{1}{2}}$, which, with Eq. (16), implies $\mathbf{A}\mathbf{A}^\top = \mathbf{M}$. Recall that $\mathcal{G}$ is a DAG, which indicates $\mathbf{B}$ also represents a DAG, Therefore, $\mathbf{I} - \mathbf{B}$, and thus $\mathbf{A}$, are non-singular. Since right multiplication of $\boldsymbol{\Omega}^{-\frac{1}{2}}$ does not affect the support of $\mathbf{I} - \mathbf{B}$, by Eqs. (15) and (16), we have

$$\text{supp}(\mathbf{A}) = \text{supp}(\mathbf{I} - \mathbf{B}) = \text{supp}(\mathbf{I}) \cup \text{supp}(\mathbf{B}) \subseteq \text{supp}(\mathbf{I}) \cup \text{supp}(\text{off}(\boldsymbol{\xi}')) = \text{supp}(\boldsymbol{\xi}').$$

Therefore, we have $\text{supp}(\mathbf{A}) \subseteq \text{supp}(\boldsymbol{\xi}')$, which, with the non-singularity of matrix $\mathbf{A}$, implies $\mathbf{M} = \mathbf{A}\mathbf{A}^\top \in \boldsymbol{\Sigma}(\boldsymbol{\xi}')$.

---

[2]We say that a support matrix $\boldsymbol{\xi}$ is non-singular if there exists non-singular matrix $\mathbf{A}$ such that $\text{supp}(\mathbf{A}) \subseteq \text{supp}(\boldsymbol{\xi})$.

**Proof of $\boldsymbol{\Sigma}(\boldsymbol{\xi}') \subseteq \boldsymbol{\Theta}(\mathcal{G})$:**

Suppose $\mathbf{M} \in \boldsymbol{\Sigma}(\boldsymbol{\xi}')$. By definition of $\boldsymbol{\Sigma}(\boldsymbol{\xi}')$, there exists non-singular matrix $\mathbf{A} \in \mathbb{R}^{n \times n}$ such that

$$\mathrm{supp}(\mathbf{A}) \subseteq \mathrm{supp}(\boldsymbol{\xi}') \quad \text{and} \quad \mathbf{M} = \mathbf{A}\mathbf{A}^\top. \tag{17}$$

Since matrix $\mathbf{A}$ is non-singular, all of its diagonal entries must be nonzero, because otherwise the corresponding determinant will be zero, which contradicts its non-singularity. By Lemma 4, there exists matrix $\mathbf{B} \in \mathbb{R}_{\mathrm{off}}^{n \times n}$ and $\boldsymbol{\Omega} \in \mathrm{diag}(\mathbb{R}_{>0}^n)$ such that $\mathbf{A} = (\mathbf{I} - \mathbf{B})\boldsymbol{\Omega}^{-\frac{1}{2}}\mathbf{D}$, where $\mathbf{D}$ is a diagonal matrix with diagonal entries being $\pm 1$. By Eq. (17), we have $(\mathbf{I} - \mathbf{B})\boldsymbol{\Omega}^{-1}(\mathbf{I} - \mathbf{B})^\top = \mathbf{M}$. Since right multiplication of $\boldsymbol{\Omega}^{-\frac{1}{2}}\mathbf{D}$ does not affect the support of $\mathbf{I} - \mathbf{B}$, by Eqs. (15) and (17), we have

$$\mathrm{supp}(\mathbf{I}) \cup \mathrm{supp}(\mathbf{B}) = \mathrm{supp}(\mathbf{I} - \mathbf{B}) = \mathrm{supp}(\mathbf{A}) \subseteq \mathrm{supp}(\boldsymbol{\xi}) = \mathrm{supp}(\mathbf{I}) \cup \mathrm{supp}(\mathrm{off}(\boldsymbol{\xi}')).$$

Note that $\mathrm{supp}(\mathbf{I})$ and $\mathrm{supp}(\mathbf{B})$ are disjoint. Furthermore, $\mathrm{supp}(\mathbf{I})$ and $\mathrm{supp}(\mathrm{off}(\boldsymbol{\xi}'))$ are disjoint. Therefore, we have $\mathrm{supp}(\mathbf{B}) \subseteq \mathrm{supp}(\mathrm{off}(\boldsymbol{\xi}'))$ and thus $\mathbf{M} = (\mathbf{I} - \mathbf{B})\boldsymbol{\Omega}^{-1}(\mathbf{I} - \mathbf{B})^\top \in \boldsymbol{\Theta}(\mathcal{G})$. $\quad\square$

**Proposition 7.** *Let $\boldsymbol{\xi}_1$ and $\boldsymbol{\xi}_2$ be non-singular support matrices that satisfy Assumption 2. If they have the same set of equality constraints, i.e., $H(\boldsymbol{\xi}_2) = H(\boldsymbol{\xi}_2)$, then they are covariance equivalent.*

*Proof.* Since matrices $\boldsymbol{\xi}_1$ and $\boldsymbol{\xi}_2$ are non-singular and satisfy Assumption 2, by Lemma 10, there exist DAGs $\mathcal{G}_1$ and $\mathcal{G}_2$ such that

$$\boldsymbol{\Sigma}(\boldsymbol{\xi}_1) = \boldsymbol{\Theta}(\mathcal{G}_1) \quad \text{and} \quad \boldsymbol{\Sigma}(\boldsymbol{\xi}_2) = \boldsymbol{\Theta}(\mathcal{G}_2), \tag{18}$$

which imply

$$H(\boldsymbol{\xi}_1) = H(\mathcal{G}_1) \quad \text{and} \quad H(\boldsymbol{\xi}_2) = H(\mathcal{G}_2). \tag{19}$$

We now provide a proof by contrapositive. Suppose that $\boldsymbol{\xi}_1$ and $\boldsymbol{\xi}_2$ are not covariance equivalent, i.e., $\boldsymbol{\Sigma}(\boldsymbol{\xi}_1) \neq \boldsymbol{\Sigma}(\boldsymbol{\xi}_2)$. By Eq. (18), we have $\boldsymbol{\Theta}(\mathcal{G}_1) \neq \boldsymbol{\Theta}(\mathcal{G}_2)$, i.e., DAGs $\mathcal{G}_1$ and $\mathcal{G}_2$ are not covariance equivalent. By Ghassami et al. [20, Proposition 1], DAGs $\mathcal{G}_1$ and $\mathcal{G}_2$ are not Markov equivalent, which indicates that they do not have the same skeleton and v-structures [43]. This implies that they lead to different sets of conditional independence constraints, which in this case correspond to different sets of polynomial equality constraints [39], i.e., $H(\mathcal{G}_1) \neq H(\mathcal{G}_2)$. By Eq. (19), we have $H(\boldsymbol{\xi}_1) \neq H(\boldsymbol{\xi}_2)$. $\quad\square$

### C.4 Identifiability of Parameters from Support

In the following, we provide a result that establish the identifiability of the parameters in mixing matrix from its support.

**Proposition 8.** *Consider two non-singular matrices $\mathbf{A}_1$ and $\mathbf{A}_2$ that satisfy Assumption 2 and that entail the same covariance matrix, i.e., $\boldsymbol{\Sigma} = \mathbf{A}_1\mathbf{A}_1^\top = \mathbf{A}_2\mathbf{A}_2^\top$. If matrices $\mathbf{A}_1$ and $\mathbf{A}_2$ have the same support, then they differ only in sign changes of columns.*

*Proof.* Since matrix $\mathbf{A}_1$ satisfies Assumption 2, there exist permutation matrices $\mathbf{P}_1$ and $\mathbf{P}_2$ such that $\mathbf{P}_1^\top \mathbf{A}_1 \mathbf{P}_2$ is lower triangular. Clearly, $\mathbf{P}_1^\top \mathbf{A}_2 \mathbf{P}_2$ is also lower triangular, because matrices $\mathbf{A}_1$ and $\mathbf{A}_2$ have the same support. Since matrices $\mathbf{P}_1^\top \mathbf{A}_1 \mathbf{P}_2$ and $\mathbf{P}_1^\top \mathbf{A}_2 \mathbf{P}_2$ are non-singular, all diagonal entries of these two matrices must be nonzero, because otherwise the corresponding determinant will be zero, which contradict their non-singularity. Let $\mathbf{D}_1$ and $\mathbf{D}_2$ be diagonal matrices with diagonal entries being $\pm 1$ such that the diagonal entries of $\mathbf{P}_1^\top \mathbf{A}_1 \mathbf{D}_1 \mathbf{P}_2$ and $\mathbf{P}_1^\top \mathbf{A}_2 \mathbf{D}_2 \mathbf{P}_2$ are positive. (The procedure for constructing such diagonal matrices $\mathbf{D}_1$ and $\mathbf{D}_2$ is straightforward and omitted here.)

Furthermore, we have

$$(\mathbf{P}_1^\top \mathbf{A}_1 \mathbf{D}_1 \mathbf{P}_2)(\mathbf{P}_1^\top \mathbf{A}_1 \mathbf{D}_1 \mathbf{P}_2)^\top = (\mathbf{P}_1^\top \mathbf{A}_2 \mathbf{D}_2 \mathbf{P}_2)(\mathbf{P}_1^\top \mathbf{A}_2 \mathbf{D}_2 \mathbf{P}_2)^\top = \mathbf{P}_1^\top \boldsymbol{\Sigma} \mathbf{P}_1. \tag{20}$$

Since matrix $\mathbf{A}_1$ is non-singular, matrices $\boldsymbol{\Sigma}$, and thus $\mathbf{P}_1^\top \boldsymbol{\Sigma} \mathbf{P}_1$, are symmetric positive definite. Here, Eq. (20) can be viewed as the Cholesky decomposition of $\mathbf{P}_1^\top \boldsymbol{\Sigma} \mathbf{P}_1$, where $\mathbf{P}_1^\top \mathbf{A}_1 \mathbf{D}_1 \mathbf{P}_2$ and $\mathbf{P}_1^\top \mathbf{A}_2 \mathbf{D}_2 \mathbf{P}_2$ are the Cholesky factors. Recall that they are lower triangular matrices with all diagonal entries being positive; in this case, it is known that such Cholesky factor is unique [42]. Therefore, we have

$$\mathbf{P}_1^\top \mathbf{A}_1 \mathbf{D}_1 \mathbf{P}_2 = \mathbf{P}_1^\top \mathbf{A}_2 \mathbf{D}_2 \mathbf{P}_2,$$

which implies

$$\mathbf{A}_1 = \mathbf{A}_2 \mathbf{D}_2 \mathbf{D}_1^{-1} = \mathbf{A}_2 \mathbf{D}_2 \mathbf{D}_1.$$

Since $\mathbf{D}_2 \mathbf{D}_1$ is a diagonal matrix with diagonal entries being $\pm 1$, we conclude that matrices $\mathbf{A}_1$ and $\mathbf{A}_2$ differ only in sign changes of columns. $\quad\square$

**Corollary 1 (Identifiability of Parameters from Support).** *Consider two non-singular matrices $\mathbf{A}_1$ and $\mathbf{A}_2$ that satisfy Assumption 2 and that entail the same covariance matrix, i.e., $\mathbf{\Sigma} = \mathbf{A}_1\mathbf{A}_1^\top = \mathbf{A}_2\mathbf{A}_2^\top$. If the columns of $\boldsymbol{\xi}_{\mathbf{A}_1}$ are a permutation of those of $\boldsymbol{\xi}_{\mathbf{A}_2}$, then we have $\mathbf{A}_1 \sim \mathbf{A}_2$.*

*Proof.* Suppose by contradiction that $\mathbf{A}_1 \not\sim \mathbf{A}_2$. This implies that, for every permutation matrix $\mathbf{P}$ such that $\boldsymbol{\xi}_{\mathbf{A}_1} = \boldsymbol{\xi}_{\mathbf{A}_2\mathbf{P}}$, matrices $\mathbf{A}_1$ and $\mathbf{A}_2\mathbf{P}$ differ in more than sign changes of columns. By Proposition 8, this cannot happen. $\qquad\square$

# D    Proofs of Other Results

## D.1    Proof of Example 1

**Example 1 (Semialgebraic Constraints).** *Consider support matrices*

$$
\boldsymbol{\xi}_1 = \begin{bmatrix} \times & 0 & 0 \\ \times & \times & 0 \\ \times & 0 & \times \end{bmatrix} \quad \text{and} \quad \boldsymbol{\xi}_2 = \begin{bmatrix} \times & 0 & \times \\ \times & \times & 0 \\ 0 & \times & \times \end{bmatrix}.
$$

*The equality constraints imposed by $\boldsymbol{\xi}_1$ include*

$$
\Sigma_{1,1}\Sigma_{2,3} - \Sigma_{1,2}\Sigma_{1,3} = 0,
$$

*while the inequality constraints imposed by $\boldsymbol{\xi}_2$ include*

$$
(\Sigma_{1,1}\Sigma_{2,2}\Sigma_{3,3}+\Sigma_{1,1}\Sigma_{2,3}^2-\Sigma_{2,2}\Sigma_{1,3}^2-\Sigma_{3,3}\Sigma_{1,2}^2)^2-4(\Sigma_{1,1}\Sigma_{2,2}-\Sigma_{1,2}^2)(\Sigma_{1,1}\Sigma_{3,3}\Sigma_{2,3}^2-\Sigma_{1,3}^2\Sigma_{2,3}^2) \geq 0.
$$

*Proof.* We first consider matrix $\mathbf{A}_1$ with the support $\boldsymbol{\xi}_1$. That is, matrix $\mathbf{A}_1$ is of the form

$$
\mathbf{A}_1 = \begin{bmatrix} a_{1,1} & 0 & 0 \\ a_{2,1} & a_{2,2} & 0 \\ a_{3,1} & 0 & a_{3,3} \end{bmatrix}.
$$

The entries of the resulting covariance matrix $\mathbf{\Sigma} = \mathbf{A}_1\mathbf{A}_1^\top$ are then given by

$$
\begin{aligned}
\Sigma_{1,1} &= a_{1,1}^2, & \Sigma_{1,2} &= a_{1,1}a_{2,1}, \\
\Sigma_{2,2} &= a_{2,1}^2 + a_{2,2}^2, & \Sigma_{1,3} &= a_{1,1}a_{3,1}, \\
\Sigma_{3,3} &= a_{3,1}^2 + a_{3,3}^2, & \Sigma_{2,3} &= a_{2,1}a_{3,1},
\end{aligned}
$$

which imply

$$
\Sigma_{1,1}\Sigma_{2,3} - \Sigma_{1,2}\Sigma_{1,3} = 0.
$$

We now consider matrix $\mathbf{A}_2$ with the support $\boldsymbol{\xi}_2$. That is, matrix $\mathbf{A}_2$ is of the form

$$
\mathbf{A}_2 = \begin{bmatrix} a_{1,1} & 0 & a_{1,3} \\ a_{2,1} & a_{2,2} & 0 \\ 0 & a_{3,2} & a_{3,3} \end{bmatrix}.
$$

The entries of the resulting covariance matrix $\mathbf{\Sigma} = \mathbf{A}_2\mathbf{A}_2^\top$ are then given by

$$
\begin{aligned}
\Sigma_{1,1} &= a_{1,1}^2 + a_{1,3}^2, & \Sigma_{1,2} &= a_{1,1}a_{2,1}, \\
\Sigma_{2,2} &= a_{2,1}^2 + a_{2,2}^2, & \Sigma_{1,3} &= a_{1,3}a_{3,3}, \\
\Sigma_{3,3} &= a_{3,2}^2 + a_{3,3}^2, & \Sigma_{2,3} &= a_{2,2}a_{3,2}.
\end{aligned}
$$

Suppose we fix the value of $a_{3,2}$. This leads to

$$
a_{3,2}^2 = \Sigma_{3,3} - \frac{\Sigma_{1,3}^2}{\Sigma_{1,1} - \frac{\Sigma_{1,2}^2}{\Sigma_{2,2} - \frac{\Sigma_{2,3}^2}{a_{3,2}^2}}},
$$

which can be rewritten as

$$
(\Sigma_{1,1}\Sigma_{2,2}-\Sigma_{1,2}^2)a_{3,2}^4+(-\Sigma_{1,1}\Sigma_{2,2}\Sigma_{3,3}-\Sigma_{1,1}\Sigma_{2,3}^2+\Sigma_{2,2}\Sigma_{1,3}^2+\Sigma_{3,3}\Sigma_{1,2}^2)a_{3,2}^2+(\Sigma_{1,1}\Sigma_{3,3}\Sigma_{2,3}^2-\Sigma_{1,3}^2\Sigma_{2,3}^2) = 0.
$$

Since the value of $a_{3,2}$ is a real number, we have

$$
(\Sigma_{1,1}\Sigma_{2,2}\Sigma_{3,3} + \Sigma_{1,1}\Sigma_{23}^2 - \Sigma_{2,2}\Sigma_{1,3}^2 - \Sigma_{3,3}\Sigma_{1,2}^2)^2 - 4(\Sigma_{1,1}\Sigma_{2,2} - \Sigma_{1,2}^2)(\Sigma_{1,1}\Sigma_{3,3}\Sigma_{2,3}^2 - \Sigma_{1,3}^2\Sigma_{2,3}^2) \geq 0.
$$

$\qquad\square$

## D.2 Proof of Proposition 1

**Proposition 1.** *If the true mixing matrix $\tilde{\mathbf{A}}$ does not satisfy Assumption 1, then there exists a solution $\hat{\mathbf{A}}$ to Problem (1) such that $\hat{\mathbf{A}} \not\sim \tilde{\mathbf{A}}$.*

*Proof.* Since the true mixing matrix $\tilde{\mathbf{A}}$ does not satisfy Assumption 1, there exist $i, j \in [n]$, $i \neq j$ such that
$$|\operatorname{supp}(\tilde{\mathbf{a}}_i) \cup \operatorname{supp}(\tilde{\mathbf{a}}_j)| - |\operatorname{supp}(\tilde{\mathbf{a}}_i) \cap \operatorname{supp}(\tilde{\mathbf{a}}_j)| \leq 1.$$
This leads to the following two cases:

- **Case 1:** $|\operatorname{supp}(\tilde{\mathbf{a}}_i) \cup \operatorname{supp}(\tilde{\mathbf{a}}_j)| - |\operatorname{supp}(\tilde{\mathbf{a}}_i) \cap \operatorname{supp}(\tilde{\mathbf{a}}_j)| = 1$. In this case, since the mixing matrix is of full column rank, there must exist a $k \in [n]$ such that $(\boldsymbol{\xi}_{\tilde{\mathbf{A}}})_{k,i} = (\boldsymbol{\xi}_{\tilde{\mathbf{A}}})_{k,j} = \times$. Thus, we can always apply a reversible acute rotation (see Remark 3) to the $i$-th and $j$-th columns of matrix $\tilde{\mathbf{A}}$. This operation leads to another matrix $\ddot{\mathbf{A}}$ with $\|\ddot{\mathbf{A}}\|_0 = \|\tilde{\mathbf{A}}\|_0$ and $\ddot{\mathbf{A}}\ddot{\mathbf{A}}^\top = \tilde{\mathbf{A}}\tilde{\mathbf{A}}^\top$. In the reversible acute rotation, we can set either $(\boldsymbol{\xi}_{\tilde{\mathbf{A}}})_{k,i}$ or $(\boldsymbol{\xi}_{\tilde{\mathbf{A}}})_{k,j}$ to 0. This implies $\|\ddot{\mathbf{a}}_{k,:}\|_0 < \|\tilde{\mathbf{a}}_{k,:}\|_0$, and therefore $\ddot{\mathbf{A}} \not\sim \tilde{\mathbf{A}}$. Now, suppose that the true mixing matrix $\tilde{\mathbf{A}}$ is not a solution to Problem (1). Clearly, there must exist a solution $\hat{\mathbf{A}}$ to Problem (1) such that $\|\hat{\mathbf{A}}\|_0 < \|\tilde{\mathbf{A}}\|_0$, which indicates $\hat{\mathbf{A}} \not\sim \tilde{\mathbf{A}}$. It remains to consider the case where matrix $\tilde{\mathbf{A}}$ is a solution to Problem (1). In this case, since $\|\ddot{\mathbf{A}}\|_0 = \|\tilde{\mathbf{A}}\|_0$, matrix $\ddot{\mathbf{A}}$ is also a solution to Problem (1), and we have shown that $\ddot{\mathbf{A}} \not\sim \tilde{\mathbf{A}}$.

- **Case 2:** $|\operatorname{supp}(\tilde{\mathbf{a}}_i) \cup \operatorname{supp}(\tilde{\mathbf{a}}_j)| - |\operatorname{supp}(\tilde{\mathbf{a}}_i) \cap \operatorname{supp}(\tilde{\mathbf{a}}_j)| = 0$. In this case, we can always apply a reduction (see Remark 3) to the $i$-th and $j$-th columns of matrix $\tilde{\mathbf{A}}$. This operation leads to another matrix $\ddot{\mathbf{A}}$ with $\|\ddot{\mathbf{A}}\|_0 < \|\tilde{\mathbf{A}}\|_0$ and $\ddot{\mathbf{A}}\ddot{\mathbf{A}}^\top = \tilde{\mathbf{A}}\tilde{\mathbf{A}}^\top$. Therefore, there must exist a solution $\hat{\mathbf{A}}$ to Problem (1) such that $\|\hat{\mathbf{A}}\|_0 \leq \|\ddot{\mathbf{A}}\|_0 < \|\tilde{\mathbf{A}}\|_0$, which indicates $\hat{\mathbf{A}} \not\sim \tilde{\mathbf{A}}$.

In either case, there exists a solution to Problem (1) whose columns are not signed permutations of the columns of $\tilde{\mathbf{A}}$. $\qquad\square$

## D.3 Proof of Example 2

**Example 2.** *If the connective structure $\mathcal{G}_{\mathbf{A}}$ of mixing matrix $\mathbf{A}$ is a polytree, then matrix $\mathbf{A}$ satisfies Assumption 2.*

*Proof.* Suppose that the matrix $\mathbf{A}$ does not satisfy Assumption 2. This means that there does not exist permutation matrices $\mathbf{P}_1$ and $\mathbf{P}_2$ such that $\mathbf{P}_1^\top \mathbf{A} \mathbf{P}_2$ is lower triangular, which implies that, in the connective structure $\mathcal{G}_{\mathbf{A}}$, there exists a path that alternates between source and observed variable nodes, i.e., a sequence of nodes $\{s_{i_1}, x_{j_1}, s_{i_2}, x_{j_2}, ..., s_{i_k}, x_{j_k}, s_{i_1}\}$ where each pair $(s_{i_t}, x_{j_t})$ corresponds to a nonzero entry $a_{j_t, i_t}$ in $\mathbf{A}$, for $t = 1, \ldots, k$. Thus, by replacing the directed edges on this path with undirected edges, we obtain a cycle. Therefore, $\mathcal{G}_{\mathbf{A}}$ cannot be a polytree. $\qquad\square$

## D.4 Proof of Proposition 3

The proof of the following proposition is adapted from that of Ghassami et al. [20, Proposition 8].

**Proposition 3 (Generic Property).** *Suppose that the nonzero coefficients of matrix $\mathbf{A}$ are randomly drawn from a distribution that is absolutely continuous with respect to Lebesgue measure. Then, matrix $\mathbf{A}$ satisfies Assumption 3 with probability one.*

*Proof.* Let $\phi$ be the set of possible equality constraints of any covariance matrix with the same size as $\boldsymbol{\Sigma}$, which is a finite set because the number of variables is finite. Consider a matrix $\hat{\mathbf{A}}$ with the same support as $\mathbf{A}$ (i.e., $\boldsymbol{\xi}_{\hat{\mathbf{A}}} = \boldsymbol{\xi}_{\mathbf{A}}$) that violates Assumption 3, where the corresponding covariance matrix is $\hat{\boldsymbol{\Sigma}}$. To violate Assumption 3, $\hat{\boldsymbol{\Sigma}}$ has to satisfy an equality constraint $\kappa \in \phi \setminus H(\boldsymbol{\xi}_{\mathbf{A}})$. Thus, the set of possible matrices with the same support as $\mathbf{A}$ that violate Assumption 3 is a subset of

$$\bigcup_{\kappa \in \phi \setminus H(\boldsymbol{\xi}_{\mathbf{A}})} \left\{ \hat{\mathbf{A}} : \boldsymbol{\xi}_{\hat{\mathbf{A}}} = \boldsymbol{\xi}_{\mathbf{A}} \text{ and } \hat{\boldsymbol{\Sigma}} \text{ satisfies equality constraint } \kappa \right\}.$$

By the definition of equality constraint, each set in the union above has zero Lebesgue measure, and thus the finite union above also has zero Lebesgue measure. This implies that the set of possible

matrices with the same support as $\mathbf{A}$ that violate Assumption 3 has zero Lebesgue measure. Therefore, Assumption 3 is satisfied with probability one. $\qquad\square$

### D.5 Proof of Theorem 2

We first prove the following proposition that will be used to prove both Theorems 2 and 3.

**Proposition 9.** *If mixing matrix $\mathbf{A}$ satisfies Assumption 6, then it satisfies Assumption 1.*

*Proof.* We provide a proof by contrapositive. Suppose $\mathbf{A}$ does not satisfy Assumption 1. This means that there exist some $i, j \in [n]$ with $i \neq j$ such that

$$|\operatorname{supp}(\mathbf{a}_i) \cup \operatorname{supp}(\mathbf{a}_j)| - |\operatorname{supp}(\mathbf{a}_i) \cap \operatorname{supp}(\mathbf{a}_j)| \leq 1.$$

The difference can be either 0 or 1.

- **Case 1:** $|\operatorname{supp}(\mathbf{a}_i) \cup \operatorname{supp}(\mathbf{a}_j)| - |\operatorname{supp}(\mathbf{a}_i) \cap \operatorname{supp}(\mathbf{a}_j)| = 0$. This implies $\operatorname{supp}(\mathbf{a}_i) = \operatorname{supp}(\mathbf{a}_j)$, i.e., the supports of $\mathbf{a}_i$ and $\mathbf{a}_j$ are identical. In this case, Assumption 6 is clearly violated, as $\operatorname{supp}(\mathbf{a}_i)$ is a subset of $\operatorname{supp}(\mathbf{a}_j)$ and vice versa.

- **Case 2:** $|\operatorname{supp}(\mathbf{a}_i) \cup \operatorname{supp}(\mathbf{a}_j)| - |\operatorname{supp}(\mathbf{a}_i) \cap \operatorname{supp}(\mathbf{a}_j)| = 1$. This implies that one of the columns is a proper subset of the other, meaning either $\operatorname{supp}(\mathbf{a}_i) \subset \operatorname{supp}(\mathbf{a}_j)$ or $\operatorname{supp}(\mathbf{a}_j) \subset \operatorname{supp}(\mathbf{a}_i)$. Again, this violates Assumption 6.

Hence, in either case, if $\mathbf{A}$ does not satisfy Assumption 1, then it does not satisfy Assumption 6. $\square$

We now prove the following theorem.

**Theorem 2.** *For mixing matrix $\mathbf{A}$, we have the following chain of chain of implications:*

$$\textit{Assumption } 4 \implies \textit{Assumption } 6 \implies \textit{Assumption } 1.$$

*Furthermore, there exists a matrix $\mathbf{A}$ satisfying Assumption 1 that does not satisfy Assumption 4.*

*Proof.* We first prove Assumption 4 $\implies$ Assumption 6.

In Assumption 4, Eq. (3) is assumed to be satsified for all $\mathcal{I} \subseteq [n]$ where $|\mathcal{I}| > 1$. Thus, in order to prove that Assumption 4 implies Assumption 6, it is sufficient to only consider the case where $|\mathcal{I}| = 2$. That is, for every $i, j \in [n]$ and $i \neq j$, we have

$$|\operatorname{supp}(\mathbf{a}_i) \cup \operatorname{supp}(\mathbf{a}_j)| > \operatorname{rank}(\operatorname{overlap}(\mathbf{A}_{\{i,j\}})) + |\operatorname{supp}(\mathbf{a}_i)|,$$

where we set $i$ as the target index without loss of generality.

Because $\operatorname{rank}(\operatorname{overlap}(\mathbf{A}_{\{i,j\}})) \geq 1$, we have

$$|\operatorname{supp}(\mathbf{a}_i) \cup \operatorname{supp}(\mathbf{a}_j)| > 1 + |\operatorname{supp}(\mathbf{a}_i)|. \tag{21}$$

Suppose $|\operatorname{supp}(\mathbf{a}_i)| \geq |\operatorname{supp}(\mathbf{a}_j)|$. Eq. (21) implies that $\operatorname{supp}(\mathbf{a}_j)$ is not a subset of $\operatorname{supp}(\mathbf{a}_i)$. Similarly, if $|\operatorname{supp}(\mathbf{a}_j)| \geq |\operatorname{supp}(\mathbf{a}_i)|$, we could also show that $\operatorname{supp}(\mathbf{a}_i)$ is not a subset of $\operatorname{supp}(\mathbf{a}_j)$. Thus, Assumption 6 is satisfied.

According to Proposition 9, we have Assumption 6 $\implies$ Assumption 1, which completes the proof of the first part, i.e., Assumption 4 $\implies$ Assumption 6 $\implies$ Assumption 1.

We now provide an example of mixing matrix satisfying the Assumption 1 that does not satisfy Assumption 4. Suppose the mixing matrix $\tilde{\mathbf{A}}$ has a support as follows:

$$\boldsymbol{\xi}_{\tilde{\mathbf{A}}} = \begin{bmatrix} \times & 0 & 0 \\ \times & \times & 0 \\ \times & 0 & \times \end{bmatrix}$$

Clearly, Assumption 1 is satisfied since every pair of columns differ on more than one entry. However, for $k = 1$ and $\mathcal{I} = \{1, 2\}$, we have

$$\left| \bigcup_{k' \in \mathcal{I}} \operatorname{supp}(\mathbf{a}_{k'}) \right| - \operatorname{rank}(\operatorname{overlap}(\mathbf{A}_{\mathcal{I}})) \leq \left| \bigcup_{k' \in \mathcal{I}} \operatorname{supp}(\mathbf{a}_{k'}) \right| = 3.$$

Since $|\operatorname{supp}(\mathbf{a}_k)| = |\operatorname{supp}(\mathbf{a}_1)| = 3$, it is not possible for Eq. (3) to hold for $\tilde{\mathbf{A}}$ when $k = 1$ and $\mathcal{I} = \{1, 2\}$. Thus, Assumption 4 is violated. The proof of part (b) is finished. $\qquad\square$

## D.6 Proof of Theorem 3

**Theorem 3.** *For mixing matrix* $\mathbf{A}$*, we have the following chain of chain of implications:*

$$\text{Assumption 5} \implies \text{Assumption 6} \implies \text{Assumption 1}.$$

*Furthermore, there exists a matrix* $\mathbf{A}$ *satisfying Assumption 1 that does not satisfy Assumption 5.*

*Proof.* We first prove the contrapositive of Assumption 5 $\implies$ Assumption 6. Suppose that Assumption 6 does not hold. This means that there exist distinct indices $i, j \in [n]$, such that $\mathrm{supp}(\mathbf{a}_i)$ is a subset of $\mathrm{supp}(\mathbf{a}_j)$.

Now, for any set of row indices $\mathcal{I} \subset [n]$, consider the intersection over the supports of all rows $j \in \mathcal{I}$, denoted by $\bigcap_{j \in \mathcal{I}} \mathrm{supp}(\mathbf{a}_{j,:})$. Because $\mathrm{supp}(\mathbf{a}_i)$ is a subset of $\mathrm{supp}(\mathbf{a}_j)$, it is clear that $i$ cannot be the only element in this intersection. Therefore, it is impossible to satisfy $\bigcap_{j \in \mathcal{I}} \mathrm{supp}(\mathbf{a}_{j,:}) = \{i\}$ for any choice of $\mathcal{I}$, which indicates that Assumption 5 is violated.

According to Proposition 9, we have Assumption 6 $\implies$ Assumption 1, which completes the proof of the first part, i.e., Assumption 5 $\implies$ Assumption 6 $\implies$ Assumption 1.

We now provide an example of mixing matrix satisfying the Assumption 1 that does not satisfy Assumption 5. Suppose the mixing matrix $\tilde{\mathbf{A}}$ has a support as follows:

$$\boldsymbol{\xi}_{\tilde{\mathbf{A}}} = \begin{bmatrix} \times & 0 & 0 \\ \times & \times & 0 \\ \times & 0 & \times \end{bmatrix}$$

Clearly, Assumption 1 is satisfied since the supports of each pair of columns differ on more than one entry. However, Assumption 5 is violated since there does not exist any set of rows such that the intersection of their nonzero indices is 2 or 3. $\qquad\square$

## D.7 Proof of Theorem 4

**Theorem 4 (Equivalent Formulations).** *Suppose* $\tilde{\mathbf{A}} = (\mathbf{I} - \tilde{\mathbf{B}})\tilde{\boldsymbol{\Omega}}^{-\frac{1}{2}}$*. Then, we have:*

(a) *Let* $(\hat{\mathbf{B}}, \hat{\boldsymbol{\Omega}})$ *be a solution to Problem* (4)*. Then,* $\hat{\mathbf{A}} := (\mathbf{I} - \hat{\mathbf{B}})\hat{\boldsymbol{\Omega}}^{-\frac{1}{2}}$ *is a solution to Problem* (1)*.*

(b) *Let* $\hat{\mathbf{A}}$ *be a solution to Problem* (1)*. Then, there exist matrices* $\hat{\mathbf{B}} \in \mathbb{R}_{\mathrm{off}}^{n \times n}$ *and* $\hat{\boldsymbol{\Omega}} \in \mathrm{diag}(\mathbb{R}_{>0}^n)$ *such that* $\hat{\mathbf{A}} \sim (\mathbf{I} - \hat{\mathbf{B}})\hat{\boldsymbol{\Omega}}^{-\frac{1}{2}}$*, and* $(\hat{\mathbf{B}}, \hat{\boldsymbol{\Omega}})$ *is a solution to Problem* (4)*.*

*Proof.* We consider both parts of the statements.

**Part (a):**

We provide a proof by contrapositive. For matrices $\hat{\mathbf{B}}$ and $\hat{\boldsymbol{\Omega}}$, suppose $\hat{\mathbf{A}} := (\mathbf{I} - \hat{\mathbf{B}})\hat{\boldsymbol{\Omega}}^{-\frac{1}{2}}$ is not a solution to Problem (1). That is, there exists matrix $\ddot{\mathbf{A}}$ such that

$$\|\ddot{\mathbf{A}}\|_0 < \|\hat{\mathbf{A}}\|_0 \tag{22}$$

and

$$\ddot{\mathbf{A}}\ddot{\mathbf{A}}^\top = \tilde{\mathbf{A}}\tilde{\mathbf{A}}^\top. \tag{23}$$

By Lemma 3, matrix $\ddot{\mathbf{A}}$ is non-singular, and thus, by Lemma 5, there exist matrices $\ddot{\mathbf{B}} \in \mathbb{R}_{\mathrm{off}}^{n \times n}$ and $\ddot{\boldsymbol{\Omega}} \in \mathrm{diag}(\mathbb{R}_{>0}^n)$ such that

$$\ddot{\mathbf{A}} \sim (\mathbf{I} - \ddot{\mathbf{B}})\ddot{\boldsymbol{\Omega}}^{-\frac{1}{2}}.$$

Lemma 6 implies

$$\|\hat{\mathbf{A}}\|_0 = \|\hat{\mathbf{B}}\|_0 + n \quad \text{and} \quad \|\ddot{\mathbf{A}}\|_0 = \|\ddot{\mathbf{B}}\|_0 + n,$$

which, with Inequality (22), indicate $\|\ddot{\mathbf{B}}\|_0 < \|\hat{\mathbf{B}}\|_0$. Furthermore, using Eq. (23) and the assumption $\tilde{\mathbf{A}} = (\mathbf{I} - \tilde{\mathbf{B}})\tilde{\boldsymbol{\Omega}}^{-\frac{1}{2}}$, we have

$$(\mathbf{I} - \ddot{\mathbf{B}})\ddot{\boldsymbol{\Omega}}^{-1}(\mathbf{I} - \ddot{\mathbf{B}})^\top = (\mathbf{I} - \tilde{\mathbf{B}})\tilde{\boldsymbol{\Omega}}^{-1}(\mathbf{I} - \tilde{\mathbf{B}})^\top.$$

Therefore, $(\ddot{\mathbf{B}}, \ddot{\boldsymbol{\Omega}})$ satisfies the constraint of Problem (4) and leads to a smaller zero norm for the objective function, and thus $(\hat{\mathbf{B}}, \hat{\boldsymbol{\Omega}})$ will never be a solution of Problem (4).

**Part (b):**

Let $\hat{\mathbf{A}}$ be a solution to Problem (1). By Lemma 3, matrix $\hat{\mathbf{A}}$ is non-singular, and thus, by Lemma 5, there exist matrices $\hat{\mathbf{B}} \in \mathbb{R}_{\text{off}}^{n \times n}$ and $\hat{\mathbf{\Omega}} \in \text{diag}(\mathbb{R}_{>0}^n)$ such that

$$\hat{\mathbf{A}} \sim (\mathbf{I} - \hat{\mathbf{B}})\hat{\mathbf{\Omega}}^{-\frac{1}{2}}.$$

It then remains to prove that $(\hat{\mathbf{B}}, \hat{\mathbf{\Omega}})$ is a solution to Problem (4), which we do so by contradiction. Suppose that $(\hat{\mathbf{B}}, \hat{\mathbf{\Omega}})$ is not a solution to Problem (4). That is, there exists solution $(\ddot{\mathbf{B}}, \ddot{\mathbf{\Omega}})$ such that

$$\|\ddot{\mathbf{B}}\|_0 < \|\hat{\mathbf{B}}\|_0 \tag{24}$$

and

$$(\mathbf{I} - \ddot{\mathbf{B}})\ddot{\mathbf{\Omega}}^{-1}(\mathbf{I} - \ddot{\mathbf{B}})^\top = (\mathbf{I} - \tilde{\mathbf{B}})\tilde{\mathbf{\Omega}}^{-1}(\mathbf{I} - \tilde{\mathbf{B}})^\top. \tag{25}$$

Define $\ddot{\mathbf{A}} := (\mathbf{I} - \ddot{\mathbf{B}})\tilde{\mathbf{\Omega}}^{-\frac{1}{2}}$. Lemma 6 implies $\|\hat{\mathbf{A}}\|_0 = \|\hat{\mathbf{B}}\|_0 + n$ and $\|\ddot{\mathbf{A}}\|_0 = \|\ddot{\mathbf{B}}\|_0 + n$, which, with Inequality (24), indicates $\|\ddot{\mathbf{A}}\|_0 < \|\hat{\mathbf{A}}\|_0$. Furthermore, using Eq. (25) and the assumption $\tilde{\mathbf{A}} = (\mathbf{I} - \tilde{\mathbf{B}})\tilde{\mathbf{\Omega}}^{-\frac{1}{2}}$, we have

$$\ddot{\mathbf{A}}\ddot{\mathbf{A}}^\top = \tilde{\mathbf{A}}\tilde{\mathbf{A}}^\top.$$

Therefore, $\ddot{\mathbf{A}}$ satisfies the constraint of Problem (1) and leads to a smaller zero norm for the objective function, and thus $\hat{\mathbf{A}}$ will never be a solution of Problem (1), which is a contradiction. □

## D.8 Proof of Theorem 5

In this section, we first provide the proofs of Propositions 10 and 11, which together straightforwardly imply Theorem 5. Before proving Proposition 10, we state the following lemma from Shimizu et al. [38] that is useful for the proof.

**Lemma 11 (Shimizu et al. [38, Lemma 1]).** *Let $\mathbf{A}$ be a lower triangular matrix with all diagonal entries being nonzero. Let $\mathbf{P}_1$ and $\mathbf{P}_2$ be two permutation matrices. Then, a permutation of rows and columns of $\mathbf{A}$, i.e., $\mathbf{P}_1^\top \mathbf{A} \mathbf{P}_2$, has only nonzero entries in the diagonal if and only if the row and column permutations are equal, i.e., $\mathbf{P}_1 = \mathbf{P}_2$.*

We now provide the proof of Proposition 10.

**Proposition 10.** *Suppose $\mathbf{A} \sim (\mathbf{I} - \mathbf{B})\mathbf{\Omega}^{-\frac{1}{2}}$ for matrices $\mathbf{A} \in \mathbb{R}^{n \times n}$, $\mathbf{B} \in \mathbb{R}_{\text{off}}^{n \times n}$, and $\mathbf{\Omega} \in \text{diag}(\mathbb{R}_{>0}^n)$. Then, $\mathbf{A}$ satisfies Assumption 2 if and only if matrix $\mathbf{B}$ represents a DAG.*

*Proof.* Without loss of generality, we consider the case in which $\mathbf{A}$ and $(\mathbf{I} - \mathbf{B})\mathbf{\Omega}^{-\frac{1}{2}}$ differ only in column permutations, instead of signed column permutations. This is because, for the latter case, there exists a diagonal matrix $\mathbf{D}$ with diagonal entries being $\pm 1$ such that $\mathbf{A}\mathbf{D}$ and $(\mathbf{I} - \mathbf{B})\mathbf{\Omega}^{-\frac{1}{2}}$ differ only in column permutations, and furthermore, $\mathbf{A}\mathbf{D}$ satisfies Assumption 2 if and only if $\mathbf{A}$ satisfies Assumption 2.

Therefore, suppose there exists a permutation matrix $\mathbf{P}_1$ such that

$$\mathbf{A} = (\mathbf{I} - \mathbf{B})\mathbf{\Omega}^{-\frac{1}{2}}\mathbf{P}_1. \tag{26}$$

We now consider both parts of the statements.

**If part:**

Suppose that matrix $\mathbf{B}$ represents a DAG. Then, there exists permutation matrix $\mathbf{P}_2$ such that $\mathbf{P}_2^\top \mathbf{B} \mathbf{P}_2$ is strictly lower triangular. Therefore, $\mathbf{P}_2^\top (\mathbf{I} - \mathbf{B})\mathbf{P}_2 = \mathbf{I} - \mathbf{P}_2^\top \mathbf{B} \mathbf{P}_2$ is lower triangular. This implies that $\mathbf{P}_2^\top (\mathbf{I} - \mathbf{B})\mathbf{\Omega}^{-\frac{1}{2}}\mathbf{P}_2$ is lower triangular because $\mathbf{\Omega}^{-\frac{1}{2}}$ is a diagonal matrix and does not affect the support. By substituting Eq. (26), $\mathbf{P}_2^\top \mathbf{A} \mathbf{P}_1^{-1} \mathbf{P}_2$ is lower triangular. Clearly, $\mathbf{P}_1^{-1}\mathbf{P}_2$ is also a permutation matrix. Therefore, matrix $\mathbf{A}$ can be permuted by row and column permutations to be lower triangular, and thus satisfies Assumption 2.

**Only if part:**

Suppose matrix $\mathbf{A}$ satisfies Assumption 2. Then, there exist permutation matrices $\mathbf{P}_2$ and $\mathbf{P}_3$ such that $\mathbf{P}_2^\top \mathbf{A} \mathbf{P}_3$ is lower triangular. Substituting Eq. (26), $\mathbf{P}_2^\top (\mathbf{I} - \mathbf{B})\mathbf{\Omega}^{-\frac{1}{2}}\mathbf{P}_1\mathbf{P}_3$ is lower triangular.

Since $\mathbf{P}_1\mathbf{P}_3$ is also permutation matrix, this indicates that $(\mathbf{I} - \mathbf{B})\mathbf{\Omega}^{-\frac{1}{2}}$, and thus $\mathbf{I} - \mathbf{B}$, satisfy Assumption 2. By Lemma 8, $\mathbf{I} - \mathbf{B}$ is non-singular, which indicates that

$$\det(\mathbf{I} - \mathbf{B}) \neq 0.$$

Note that

$$\det(\mathbf{\Omega}^{-\frac{1}{2}}) > 0 \quad \text{and} \quad \det(\mathbf{P}_1) = 1.$$

With Eq. (26), we have

$$\det(\mathbf{A}) \neq 0,$$

and therefore

$$\det(\mathbf{P}_2^\top \mathbf{A}\mathbf{P}_3) \neq 0.$$

It is known that the determinant of a lower triangular matrix is the product of its diagonal entries. Since $\mathbf{P}_2^\top \mathbf{A}\mathbf{P}_3$ is lower triangular and $\det(\mathbf{P}_2^\top \mathbf{A}\mathbf{P}_3) \neq 0$, all diagonal entries of $\mathbf{P}_2^\top \mathbf{A}\mathbf{P}_3$ must be nonzero. Now define

$$\mathbf{P}_4 \coloneqq \mathbf{P}_2^{-1} \quad \text{and} \quad \mathbf{P}_5 \coloneqq \mathbf{P}_3^{-1}\mathbf{P}_1^{-1}, \tag{27}$$

both of which are permutation matrices. By some algebraic manipulations of Eq. (26) and further substituting the above definitions, we have

$$\mathbf{P}_4^\top(\mathbf{P}_2^\top \mathbf{A}\mathbf{P}_3)\mathbf{P}_5 = (\mathbf{I} - \mathbf{B})\mathbf{\Omega}^{-\frac{1}{2}},$$

where all diagonal entries are nonzeros. Applying Lemma 11 w.r.t. matrix $\mathbf{P}_2^\top \mathbf{A}\mathbf{P}_3$, we have

$$\mathbf{P}_4 = \mathbf{P}_5,$$

which, by plugging into Eq. (27), implies

$$\mathbf{P}_2 = \mathbf{P}_1\mathbf{P}_3.$$

Since we have shown that $\mathbf{P}_2^\top(\mathbf{I} - \mathbf{B})\mathbf{\Omega}^{-\frac{1}{2}}\mathbf{P}_1\mathbf{P}_3$ is lower triangular, further substitution of $\mathbf{P}_2 = \mathbf{P}_1\mathbf{P}_3$ indicates that $\mathbf{P}_2^\top(\mathbf{I} - \mathbf{B})\mathbf{\Omega}^{-\frac{1}{2}}\mathbf{P}_2$ is lower triangular. Since right multiplication of $\mathbf{\Omega}^{-\frac{1}{2}}$ does not affect the support of $\mathbf{I} - \mathbf{B}$, we see that

$$\mathbf{P}_2^\top(\mathbf{I} - \mathbf{B})\mathbf{P}_2 = \mathbf{I} - \mathbf{P}_2^\top \mathbf{B}\mathbf{P}_2$$

is also lower triangular. This indicates that $\mathbf{P}_2^\top \mathbf{B}\mathbf{P}_2$ is lower triangular, which, with the assumption that the diagonal entries of $\mathbf{B}$ are zeros, imply that $\mathbf{P}_2^\top \mathbf{B}\mathbf{P}_2$ is strictly lower triangular. Therefore, matrix $\mathbf{B}$ represents a DAG. □

After proving Proposition 10, we now consider Proposition 11. Before that, we provide a result by Ghassami et al. [20] that is useful for the proof. We first describe the notion of parent exchange by Ghassami et al. [20]. Let $\triangle$ be the symmetric difference operator that identifies the elements present in either of the sets but not in the intersection. For DAG $\mathcal{G}$ with weighted adjacency matrix $\mathbf{B}$, its vertices $x_i$ and $x_j$ are said to be *parent exchangeable* if $|\operatorname{supp}((\mathbf{I} - \mathbf{B})_i)\triangle \operatorname{supp}((\mathbf{I} - \mathbf{B})_j)| = 1$, i.e., there exists $k \in [n]$ such that $\operatorname{supp}((\mathbf{I} - \mathbf{B})_i)\triangle \operatorname{supp}((\mathbf{I} - \mathbf{B})_j) = \{k\}$. In such case, a support rotation can be performed on columns $(\boldsymbol{\xi}_{\mathbf{I}-\mathbf{B}})_i$ and $(\boldsymbol{\xi}_{\mathbf{I}-\mathbf{B}})_j$ that sets a nonzero entry on those columns, except $(\boldsymbol{\xi}_{\mathbf{I}-\mathbf{B}})_{i,i}$ and $(\boldsymbol{\xi}_{\mathbf{I}-\mathbf{B}})_{j,j}$, to zero. In other words, the parent of $x_i$ and $x_j$ that corresponds to the zeroed entry is removed. Furthermore, the entry $(\boldsymbol{\xi}_{\mathbf{I}-\mathbf{B}})_{k,i}$ or $(\boldsymbol{\xi}_{\mathbf{I}-\mathbf{B}})_{k,j}$ is set to $\times$, which corresponds to adding the missing edge $x_k \to x_i$ or $x_k \to x_j$. Ghassami et al. [20] defined such an operation to be a *parent exchange*. We then provide the following corollary that is straightforwardly derived from Ghassami et al. [20, Corollary 2 & Proposition 1].

**Corollary 2 (Ghassami et al. [20, Corollary 2]).** *DAGs $\mathcal{G}_1$ and $\mathcal{G}_2$ are Markov equivalent if and only if there exists a sequence of parent exchanges that maps $\mathcal{G}_1$ to $\mathcal{G}_2$, and one that maps $\mathcal{G}_2$ to $\mathcal{G}_1$.*

We now provide the proof of Proposition 11.

**Proposition 11.** *Let $\mathcal{G}$ be a DAG with weighted adjacency matrix $\mathbf{B}$. Then, matrix $\mathbf{I} - \mathbf{B}$ satisfies Assumption 1 if and only if the Markov equivalence class of $\mathcal{G}$ is a singleton.*

*Proof.* We consider both parts of the statements.

**If part:**

We provide a proof by contrapositive. Suppose that matrix $\mathbf{I} - \mathbf{B}$ does not satisfy Assumption 2. That is, there exist $i, j \in [n]$ and $i \neq j$ such that

$$|\operatorname{supp}((\mathbf{I} - \mathbf{B})_i) \triangle \operatorname{supp}((\mathbf{I} - \mathbf{B})_j)| < 1.$$

Clearly, we have $\operatorname{supp}((\mathbf{I} - \mathbf{B})_i) \neq \operatorname{supp}((\mathbf{I} - \mathbf{B})_j)$, because otherwise there will be a cycle with length of two over variables $x_i$ and $x_j$, which contradicts the assumption that $\mathcal{G}$ is a acyclic. This implies

$$|\operatorname{supp}((\mathbf{I} - \mathbf{B})_i) \triangle \operatorname{supp}((\mathbf{I} - \mathbf{B})_j)| = 1. \tag{28}$$

By definition, $x_i$ and $x_j$ are parent exchangeable, and therefore there exists a parent exchange that maps DAG $\mathcal{G}$ to another directed graph $\mathcal{G}'$ where $\mathcal{G}' \neq \mathcal{G}$. We now show by contradiction that $\mathcal{G}'$ is a DAG. Suppose that $\mathcal{G}'$ is a not DAG. By Eq. (28), there exists variable $x_k$ such that

$$\operatorname{supp}((\mathbf{I} - \mathbf{B})_i) \triangle \operatorname{supp}((\mathbf{I} - \mathbf{B})_j) = \{k\}.$$

Here, we must have $k = i$ or $k = j$, because otherwise we have

$$(\boldsymbol{\xi}_{\mathbf{I} - \mathbf{B}})_{i,i} = (\boldsymbol{\xi}_{\mathbf{I} - \mathbf{B}})_{j,j} = (\boldsymbol{\xi}_{\mathbf{I} - \mathbf{B}})_{i,j} = (\boldsymbol{\xi}_{\mathbf{I} - \mathbf{B}})_{j,i} = \times,$$

which leads to a cycle over variables $x_i$ and $x_j$. Without loss of generality, we consider the case of $k = j$, which implies

$$\operatorname{supp}((\mathbf{I} - \mathbf{B})_i) \triangle \operatorname{supp}((\mathbf{I} - \mathbf{B})_j) = \{j\} \tag{29}$$

and $(\boldsymbol{\xi}_{\mathbf{I} - \mathbf{B}})_{j,i} = 0$. This indicates that the entry $(\boldsymbol{\xi}_{\mathbf{I} - \mathbf{B}})_{j,i}$ is set to $\times$ (i.e., the edge $x_j \rightarrow x_i$ is added) after the parent exchange to DAG $\mathcal{G}'$, which subsequently leads to a cycle in $\mathcal{G}'$. In this case, there must exist a path $x_i \rightarrow x_{l_1} \rightarrow \cdots \rightarrow x_{l_m} \rightarrow x_j$ in DAG $\mathcal{G}$, where $m < n - 2$ and $l_1, \ldots, l_m \in [n] \setminus \{i, j\}$, to which adding the edge $x_j \rightarrow x_i$ leads to a cycle in DAG $\mathcal{G}'$. By Eq. (29), $x_{l_m}$ is also a parent of $x_i$, indicating that there exists a cycle $x_i \rightarrow x_{l_1} \rightarrow \cdots \rightarrow x_{l_m} \rightarrow x_i$ in DAG $\mathcal{G}$, which contradicts the assumption that $\mathcal{G}$ is acyclic. Therefore, $\mathcal{G}'$ must be a DAG.

Since there exists a parent exchange that maps DAG $\mathcal{G}$ to another DAG $\mathcal{G}'$, clearly there also exists a (reversed) parent exchange that maps DAG $\mathcal{G}'$ back to DAG $\mathcal{G}$. By Corollary 2, DAGs $\mathcal{G}$ and $\mathcal{G}'$ are Markov equivalent. Therefore, the Markov equivalence class of $\mathcal{G}$ contains at least two DAGs and is not a singleton.

**Only if part:**

Suppose that matrix $\mathbf{I} - \mathbf{B}$ satisfies Assumption 2. That is, for all $i, j \in [n]$ and $i \neq j$, we have

$$|\operatorname{supp}((\mathbf{I} - \mathbf{B})_i) \triangle \operatorname{supp}((\mathbf{I} - \mathbf{B})_j)| > 1.$$

In this case, every pair of vertices are not parent exchangeable, and thus parent exchange cannot be applied for any pair of vertices in DAG $\mathcal{G}$. Therefore, for any DAG $\mathcal{G}' \neq \mathcal{G}$, there exists no sequence of parent exchanges that maps $\mathcal{G}$ to $\mathcal{G}'$, implying that they are not Markov equivalent. This indicates that all DAGs are not Markov equivalent to DAG $\mathcal{G}$, except itself, and thus the Markov equivalence class of $\mathcal{G}$ is a singleton. $\qquad \square$

With Propositions 10 and 11 in place, we provide the proof of Theorem 5.

**Theorem 5.** *Suppose* $\mathbf{A} \sim (\mathbf{I} - \mathbf{B})\boldsymbol{\Omega}^{-\frac{1}{2}}$ *for matrices* $\mathbf{A} \in \mathbb{R}^{n \times n}$, $\mathbf{B} \in \mathbb{R}_{\text{off}}^{n \times n}$, *and* $\boldsymbol{\Omega} \in \operatorname{diag}(\mathbb{R}_{>0}^n)$. *Then,* $\mathbf{A}$ *satisfies Assumptions 1 and 2 if and only if* $\mathbf{B}$ *represents a DAG whose Markov equivalence class is a singleton.*

*Proof.* We consider both parts of the statements.

**If part:**

Suppose matrix $\mathbf{B}$ represents a DAG whose Markov equivalence class is a singleton. By Proposition 10, matrix $\mathbf{A}$ satisfies Assumption 2. Furthermore, Proposition 11 implies that $\mathbf{I} - \mathbf{B}$, and thus $(\mathbf{I} - \mathbf{B})\boldsymbol{\Omega}^{-\frac{1}{2}}$, satisfy Assumption 1. Since $\mathbf{A}$ and $(\mathbf{I} - \mathbf{B})\boldsymbol{\Omega}^{-\frac{1}{2}}$ differ only in signed column permutations, and Assumption 1 involves only pairwise comparison of the support matrix, $\mathbf{A}$ must also satisfy Assumption 1.

**Only if part:**

Suppose $\mathbf{A}$ satisfies Assumptions 1 and 2. By Proposition 10, matrix $\mathbf{B}$ represents a DAG. Since $\mathbf{A}$ and $(\mathbf{I} - \mathbf{B})\boldsymbol{\Omega}^{-\frac{1}{2}}$ differ only in signed column permutations, and Assumption 1 involves only pairwise comparison of the support matrix, $\mathbf{I} - \mathbf{B}$ also satisfies Assumption 1. By Proposition 11, the Markov equivalence class of the DAG represented by $\mathbf{B}$ is a singleton. $\qquad \square$

### D.9 Proof of Lemma 1

Before proving Lemma 1, we first provide another result that is useful for the proof. To ease further reasoning, we define the function

$$f(\mathbf{A}) = \mathrm{tr}\left(\sum_{k=1}^{n}(\mathbf{A} \odot \mathbf{A})^k\right),$$

and clearly we have

$$g(\mathbf{A}) \equiv f(\mathrm{off}(\mathbf{A})). \tag{30}$$

Zheng et al. [48], Zhang et al. [47] have shown that, for any matrix $\mathbf{A}$, $f(\mathbf{A}) = 0$ if and only if $\mathbf{A}$ represents the weighted adjacency matrix of a DAG. Also, it is known that the weighted adjacency matrix of a directed graph can be permuted via simultaneous equal row and column permutations to be strictly lower triangular if and only if the graph is a DAG. Therefore, we provide the following corollary that is straightforwardly implied by the results by Zheng et al. [48], Wei et al. [45].

**Lemma 12 (Wei et al. [45, Theorem 1]).** *For any matrix $\mathbf{A}$, $f(\mathbf{A}) = 0$ if and only if it can be permuted via simultaneous equal row and column permutations to be strictly lower triangular.*

We now provide the proof for Lemma 1.

**Lemma 1.** *For any matrix $\mathbf{A}$, $g(\mathbf{A}) = 0$ if and only if it can be permuted via simultaneous equal row and column permutations to be lower triangular.*

*Proof.* We first define $\mathbf{D_A}$ as a diagonal matrix of the same size as matrix $\mathbf{A}$, where its diagonal entries are equal to those of $\mathbf{A}$ and its non-diagonal entries are zero. Clearly, we have $\mathbf{A} = \mathrm{off}(\mathbf{A}) + \mathbf{D_A}$. For any matrix $\mathbf{P}$, This implies

$$\mathbf{P}^\top \mathbf{A} \mathbf{P} = \mathbf{P}^\top \mathrm{off}(\mathbf{A})\mathbf{P} + \mathbf{P}^\top \mathbf{D_A} \mathbf{P}. \tag{31}$$

**If part:**

Suppose that there exists permutation matrix $\mathbf{P}$ such that $\mathbf{P}^\top \mathbf{A} \mathbf{P}$ is lower triangular. By Eq. (31), we have $\mathbf{P}^\top \mathrm{off}(\mathbf{A})\mathbf{P} = \mathbf{P}^\top \mathbf{A} \mathbf{P} - \mathbf{P}^\top \mathbf{D_A} \mathbf{P}$. Clearly, the diagonal entries of $\mathbf{P}^\top \mathbf{A} \mathbf{P}$ are exactly the same as those of $\mathbf{P}^\top \mathbf{D_A} \mathbf{P}$, which cancel out each other. Therefore, the diagonal entries of $\mathbf{P}^\top \mathrm{off}(\mathbf{A})\mathbf{P}$ are zeros, indicating that it is strictly lower triangular. By Lemma 12, this implies $f(\mathrm{off}(\mathbf{A})) = 0$, and thus $g(\mathbf{A}) = 0$ by Eq. (30).

**Only if part:**

Suppose $g(\mathbf{A}) = f(\mathrm{off}(\mathbf{A})) = 0$. By Lemma 12, there exists permutation matrix $\mathbf{P}$ such that $\mathbf{P}^\top \mathrm{off}(\mathbf{A})\mathbf{P}$ is strictly lower triangular. Clearly, $\mathbf{P}^\top \mathbf{D_A} \mathbf{P}$ is a diagonal matrix. By Eq. (31), $\mathbf{P}^\top \mathbf{A} \mathbf{P}$ is lower triangular, i.e., $\mathbf{A}$ can be permuted via simultaneous equal row and column permutations to be lower triangular. $\square$

### D.10 Proof of Proposition 5

**Proposition 5.** *The matrix $\mathbf{A}$ satisfies Assumption 2 if and only if there is a matrix $\hat{\mathbf{A}}$ such that it is a column permutation of $\mathbf{A}$ and that $g(\hat{\mathbf{A}}) = 0$.*

*Proof.* We consider both parts of the statements.

**If part:**

Suppose that there exists matrix $\hat{\mathbf{A}}$ such that it is a column permutation of $\mathbf{A}$ and that $g(\hat{\mathbf{A}}) = 0$. By definition, there exists permutation matrix $\mathbf{P}_2$ such that $\hat{\mathbf{A}} = \mathbf{A}\mathbf{P}_2$. Also, by Lemma 1, there exists permutation matrix $\mathbf{P}_1$ such that $\mathbf{P}_1^\top \hat{\mathbf{A}} \mathbf{P}_1$ is lower triangular, which implies that $\mathbf{P}_1^\top \mathbf{A}\mathbf{P}_2\mathbf{P}_1$ is lower triangular. Clearly, $\mathbf{P}_2\mathbf{P}_1$ is also a permutation matrix. Therefore, matrix $\mathbf{A}$ can be permuted by row and column permutations to be lower triangular, indicating that it satisfies Assumption 2.

**Only if part:**

Suppose that matrix $\mathbf{A}$ satisfies Assumption 2, i.e., there exist permutation matrices $\mathbf{P}_1$ and $\mathbf{P}_2$ such that $\mathbf{P}_1^\top \mathbf{A}\mathbf{P}_2$ is lower triangular. Defining $\mathbf{P}_3 := \mathbf{P}_2\mathbf{P}_1^{-1}$, which is also a permutation matrix,

and substituting it into the previous statement, we see that $\mathbf{P}_1^\top \mathbf{A}\mathbf{P}_3\mathbf{P}_1$ is lower triangular. Further substitution of $\hat{\mathbf{A}} := \mathbf{A}\mathbf{P}_3$ implies that $\mathbf{P}_1^\top \hat{\mathbf{A}}\mathbf{P}_1$ is lower triangular, which, by Lemma 1, indicates $g(\hat{\mathbf{A}}) = 0$. Clearly, $\hat{\mathbf{A}}$ is a column permutation of $\mathbf{A}$. $\qquad\square$

### D.11 Proof of Theorem 6

**Theorem 6 (Alternative Formulation of Identifiability).** *Suppose that the true mixing matrix $\tilde{\mathbf{A}}$ satisfies Assumptions 1, 2, and 3. Let $\hat{\mathbf{A}}$ be a solution of the following problem:*

$$\min_{\mathbf{A}\in\mathbb{R}^{n\times n}} \|\mathbf{A}\|_0 \quad \text{subject to} \quad \mathbf{A}\mathbf{A}^\top = \tilde{\mathbf{A}}\tilde{\mathbf{A}}^\top \quad \text{and} \quad g(\mathbf{A}) = 0. \tag{5}$$

*Then, we have $\hat{\mathbf{A}} \sim \tilde{\mathbf{A}}$.*

*Proof.* Let $\hat{\mathbf{A}}$ be a solution to Problem (5). Suppose by contradiction that it is not a solution to Problem (2). That is, there exists matrix $\ddot{\mathbf{A}}$ satisfying Assumption 2 such that

$$\|\ddot{\mathbf{A}}\|_0 < \|\hat{\mathbf{A}}\|_0 \quad \text{and} \quad \ddot{\mathbf{A}}\ddot{\mathbf{A}}^\top = \tilde{\mathbf{A}}\tilde{\mathbf{A}}^\top. \tag{32}$$

By Proposition 5, there exists permutation matrix $\mathbf{P}$ such that $g(\ddot{\mathbf{A}}\mathbf{P}) = 0$. Furthermore, by Eq. (32), we have

$$\|\ddot{\mathbf{A}}\mathbf{P}\|_0 < \|\hat{\mathbf{A}}\|_0 \quad \text{and} \quad (\ddot{\mathbf{A}}\mathbf{P})(\ddot{\mathbf{A}}\mathbf{P})^\top = \tilde{\mathbf{A}}\tilde{\mathbf{A}}^\top.$$

Therefore, $\ddot{\mathbf{A}}\mathbf{P}$ satisfies the constraint of Problem (5) and leads to a smaller zero norm for the objective function, and thus $\hat{\mathbf{A}}$ will never be a solution of Problem (5), which is a contradiction. Therefore, $\hat{\mathbf{A}}$ must be a solution to Problem (2). Since matrix $\tilde{\mathbf{A}}$ also satisfies Assumptions 1, 2, and 3, applying Theorem 1 completes the proof. $\qquad\square$

### D.12 Proof of Theorem 7

**Theorem 7 (Likelihood-Based Method).** *Suppose that the true mixing matrix $\tilde{\mathbf{A}}$ satisfies Assumptions 1, 2, and 3. Let $\hat{\mathbf{A}}$ be a solution of Problem (7) with sparsity regularizer $\rho(\mathbf{A}) = 0.5\|\mathbf{A}\|_0 \log T$. Then, we have $\hat{\mathbf{A}} \sim \tilde{\mathbf{A}}$ in the large sample limit.*

*Proof.* First, we have $g(\hat{\mathbf{A}}) = 0$ and, in the large sample limit, $\bar{\mathbf{\Sigma}} = \tilde{\mathbf{\Sigma}}$. Similar to BIC [36], the likelihood term dominates in the large sample limit as the weight of the likelihood function increases much faster than that of the sparsity regularizer. Therefore, in the large sample limit, we have $\hat{\mathbf{A}}\hat{\mathbf{A}} = \tilde{\mathbf{\Sigma}} = \tilde{\mathbf{A}}\tilde{\mathbf{A}}$. Also, for any matrix $\mathbf{A}$ that satisfies $\mathbf{A}\mathbf{A} = \tilde{\mathbf{A}}\tilde{\mathbf{A}}$ and $g(\mathbf{A}) = 0$, the sparsity regularizer indicates $\|\hat{\mathbf{A}}\|_0 \le \|\mathbf{A}\|_0$, because otherwise $\hat{\mathbf{A}}$ will never be a solution of Problem (7). This implies that $\hat{\mathbf{A}}$ is also a solution to Problem (5). Since matrix $\tilde{\mathbf{A}}$ also satisfies Assumptions 1, 2, and 3, applying Theorem 1 completes the proof. $\qquad\square$

## E    Supplementary Discussion on Structural Assumptions

### E.1    Examples Satisfying Structural Variability Assumption

To illustrate the intuition of the proposed assumption of structural variability (i.e., Assumption 1), we provide several examples on the connective structure from sources to observed variables (which corresponds to the support of mixing matrix) satisfying that assumption, as illustrated in Figure 3.

### E.2    Efficient Approach for Verifying Assumption 2

Assumption 2 involves finding a certain combination of row and column permutations for mixing matrix $\mathbf{A}$, which may at first appear inefficient to verify. We provide a more efficient way to do so, by leveraging the interpretation of our assumptions in the context of causal discovery (see Section 3.4 for detailed discussion). Specifically, we provide the following corollary that is a straightforward consequence of Lemma 2 and Proposition 10, whose proof is omitted.

**Corollary 3 (Verification of Assumption 2).** *Let $\mathbf{A}$ be a non-singular matrix and $\mathbf{P}$ be a permutation matrix such that the diagonal entries of $\mathbf{A}\mathbf{P}$ are nonzero. Let $\mathcal{G} = (\mathcal{V}, \mathcal{E})$ be a directed graph where $\mathcal{V} = \{v_1, \ldots, v_n\}$ and $\mathcal{E} = \{v_j \to v_i : (\mathbf{A}\mathbf{P})_{i,j} \ne 0, i \ne j\}$. Then, directed graph $\mathcal{G}$ is acyclic if and only if matrix $\mathbf{A}$ satisfies Assumption 2.*

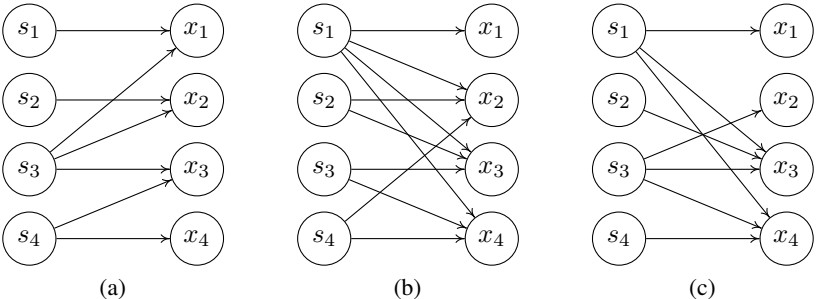

Figure 3: Graphical representations of examples that satisfy Assumption 1.

Specifically, Corollary 3 implies that it suffices to find a column permutation such that the diagonal entries of the permuted matrix are nonzero. Note that such column permutation is guaranteed to exist as indicated by Lemma 2. We then construct a directed graph $\mathcal{G}$ based on such permuted matrix and check if $\mathcal{G}$ contains cycle, e.g., via depth-first search. Instead of searching for a certain combination of row and column permutations, this procedure may be more efficient because it involves only finding specific column permutations.

## F  Supplementary Estimation Details

We provide the estimation details for the methods described in Section 4.2. In our experiments, we use the average log-likelihood $\frac{1}{T}L(\mathbf{A}; \bar{\mathbf{\Sigma}})$ as the objective (instead of $L(\mathbf{A}; \bar{\mathbf{\Sigma}})$ in Eq. (7)) for likelihood-based method. For the sparsity term $\rho(\mathbf{A})$, we use MCP with hyperparameters $\lambda = 1, \alpha = 40$ and $\lambda = 0.1, \alpha = 10$ for decomposition-based and likelihood-based methods, respectively.

Furthermore, for both methods, we use the L-BFGS algorithm [11] implemented in SciPy [44] to solve each unconstrained optimization problem of quadartic penalty method. Since the formulations involve solving nonconvex optimization problems, we run L-BFGS with 30 random initializations, where each entry of the initial solution $\mathbf{A}_0$ is sampled uniformly at random from $[-0.1, 0.1]$. In this case, the final solution is chosen via model selection. For quadratic penalty method, we use $c_1 = 10^{-5}$ and $c_1 = 10^{-2}$ for decomposition-based and likelihood-based methods, respectively, and use $\beta = 1.5$ for both methods.

Lastly, we also use a threshold of 0.01 to remove small weights in the estimated mixing matrix. We run each of the experiments on 12 CPUs and 8 GBs of memory.

**Computational complexity.** To compute the constraint term $g(\mathbf{A})$, a straightforward approach is to compute each matrix power in $g(\mathbf{A})$ and then sum their traces up, which requires $O(n)$ matrix multiplications. In our implementation, we adopt a more efficient approach with computational complexity of $O(\log n)$ matrix multiplications, inspired by Zhang et al. (2020). The rough idea is to perform exponentiation by squaring (i.e., a procedure similar to binary search) and recursively compute the term $g(\mathbf{A})$.

Furthermore, each L-BFGS run has a computational complexity of $O(m^2n^2 + m^3 + mn^2t)$, where $m \ll n^2$ is the memory size and $t$ is the number of inner iterations of the L-BFGS run. Typically, we have $t = 250$ for each L-BFGS run, and 125 iterations for the quadratic penalty method.

## G  Supplementary Experimental Results

In addition to the MCC reported in Section 5, we report the Amari distance to evaluate the identification performance in Figures 4 and 5, respectively.

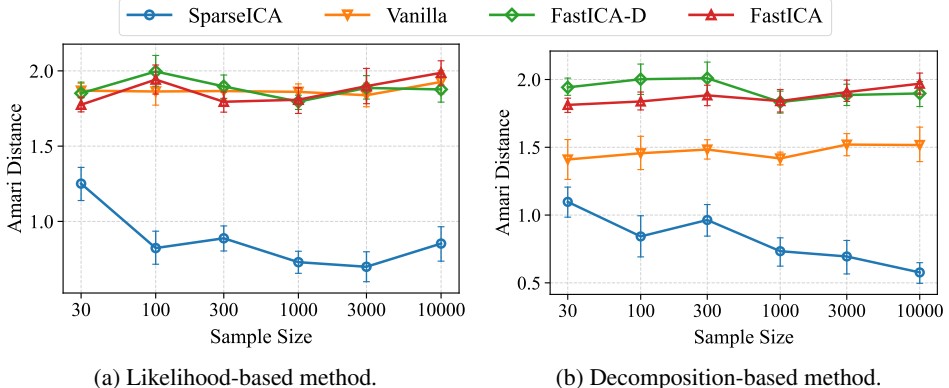

(a) Likelihood-based method.

(b) Decomposition-based method.

Figure 4: Empirical results of Amari distance across different sample sizes. Error bars indicate the standard errors calculated based on 10 random trials.

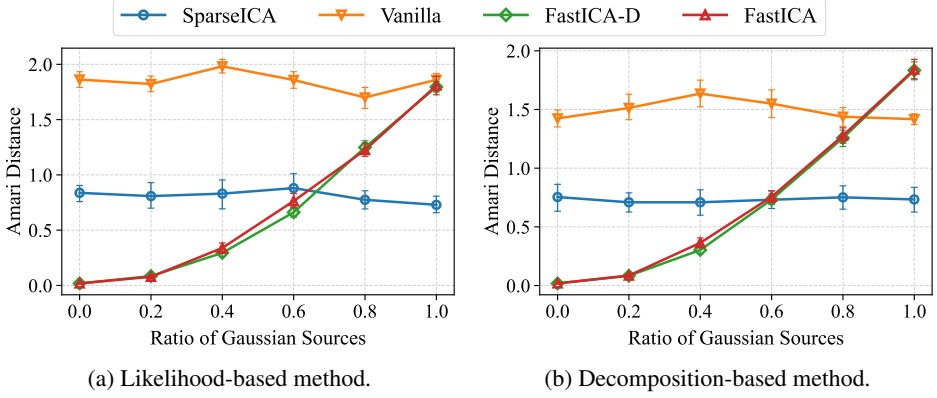

(a) Likelihood-based method.

(b) Decomposition-based method.

Figure 5: Empirical results of Amari distance across different ratios of Gaussian sources. Error bars indicate the standard errors calculated based on 10 random trials.

