# OpenReview forum: "On the Identifiability of Sparse ICA without Assuming Non-Gaussianity"
_NeurIPS.cc/2023/Conference — NeurIPS 2023 poster_

### Official Review · Reviewer_6AWD · 2023-07-06

**Soundness:** 3 good
**Presentation:** 2 fair
**Contribution:** 2 fair
**Rating:** 6
**Confidence:** 3

**Summary:**

Aiming to address rotational invariance of Gaussian sources, this work first proposed an ICA identifiability theory based on Structural Variability. It then proposed two methods based on sparsity regularization and continuous constrained optimization to estimate the mixing matrix. It also made connections between ICA and causal discovery. It finally showed preliminary results to validate the proposed theory and estimation methods.

**Strengths:**

1. The proposed method could be potentially useful as it claims to identify linear Gaussian sources with a weaker assumption compared to structural sparsity.

2. The notations, theorems and proofs are clear in general.

3. The proofs or explanations in the Appendix are very detailed and helpful.

4. The author(s) conducted experiments to validate their theory and showed the effectiveness of the proposed SparseICA method in two simulated datasets.

**Weaknesses:**

1. As the author(s) pointed out, my main concern is that there is no sufficient empirical evaluation to demonstrate the effectiveness and generalizability of the proposed theory. Current experiments only used simulated data in two settings. How does the proposed theory work on real-world datasets? This work would be more convincing if there were experiments on real-world datasets.

2. Additionally, the author(s) argued that the theory proposed by Zheng et al. 2022 couldn't capture hierarchical structures (lines 39 - 41), but it doesn't seem that the authors performed experiments to identify Gaussian sources which are organized in a hierarchical manner.

3. I don't feel all theorems/assumptions/examples were explained very clearly, but I also note that the author(s) provided detailed proofs or explanations for each proposed theorem/assumption/example in the Appendix. So I would suggest that the author(s) try to organize the manuscript more logically and guide readers to the corresponding Appendix section in the main text.

4. There are a lot of typos in the present manuscript. For example, line 71: $j$-th column by $a_{:, j}$; line 104: Section 3.2; line 248: there exists; line 254: repeated "with". Please proof-read the manuscript prior to submission.

5. There is no statistical test to compare results.

6. There is no code provided to replicate the results. Please consider making the code publicly available.

**Questions:**

1. Example 1: I understand the derivation in this example but I wonder how to identify such constraints. Do we have to derive these hard or non-hard constraints for each matrix? Are there any general principles to derive such constraints? If so, please briefly explain the principles. If not, it doesn't seem to be computationally efficient to derive these constraints as the dimension of A increases.

2. Line 130: Please elaborate what "sufficiently diverse effect" means exactly.

3. Do you have any explanations about the performance difference from two estimation methods in Figure 1? For example, decomposition method seems worse than likelihood method in vanilla case.

**Limitations:**

See Weaknesses.

---

> ### Author Rebuttal · Authors · 2023-08-09
>
> We sincerely thank the reviewer for the time dedicated to reviewing our paper and the constructive suggestions. Our responses to these comments are given below.
>
> **Q1: "no sufficient empirical evaluation" and "how does the proposed theory work on real-world datasets?"**
>
> A1: See our response to Q2 in the general response.
>
> **Q2: "it doesn't seem that the authors performed experiments to identify Gaussian sources which are organized in a hierarchical manner".**
>
> A2: Thank you for pointing this out. Given your comment, we will remove that part in Lines 39-41 to avoid possible misunderstanding. This is because we are focusing on the ICA task that is not directly related to hierarchical structures (though indirect connection may exist).
>
> As indicated by Theorems 2 and 3, the structural conditions by Zheng et al. (2020) are somewhat restrictive because they cannot handle cases where the set of observed variables influenced by one source is a superset of those affected by another source. Our proposed approach, on the other hand, is considerably less restrictive and introduces more flexibility, thereby enhancing the applicability with Gaussian sources. We will further clarify these points in the revised version to improve the clarity of our work.
>
> **Q3: "organize the manuscript more logically and guide readers to the corresponding Appendix section in the main text".**
>
> A3: Thanks for the helpful suggestion which helps improve the presentation of the paper. In the revision, we will include the 'links' in the main text to guide readers to the relevant Appendix section, especially for the proofs. Furthermore, we will organize the manuscript, such as the appendices, more logically to improve the presentation.
>
> **Q4: Typos.**
>
> A4: Thank you for your careful reading. We will carefully proofread the manuscript to fix the typos in the revision.
>
> **Q5: "There is no statistical test to compare results."**
>
> A5: Thanks for pointing this out. In light of your comment, we have additionally applied Wilcoxon signed-rank test (at $5$% significance level), and found that the improvements of the proposed method, e.g., in Figure 1, are statistically significant. We will include these statistical tests in the final version of the paper.
>
> **Q6: "There is no code provided to replicate the results. Please consider making the code publicly available."**
>
> A6: Thanks for this comment. The link to the code has been provided in Lines 862-863 in Appendix D. In light of your comment, we will move the link to Section 5 in the revised paper.
>
> **Q7: "Do we have to derive these hard or non-hard constraints for each matrix? Are there any general principles to derive such constraints?"**
>
> A7: We sincerely appreciate this insightful question. The examples of hard and non-hard constraints (Example 1) serve as illustration purposes for different types of constraints. In practice, during estimation (i.e., Algorithms 1 and 2), we do not have to derive any of these hard and non-hard constraints. The only place involving these constraints is Assumption 3, which requires that the hard constraints (if exist) of the covariance matrix arise from the support of the mixing matrix; even in this case, we do not have to derive these hard constraints. In the revision, we will include a discussion in Section 3 to make this clear.
>
> **Q8: "Please elaborate what 'sufficiently diverse effect' means exactly."**
>
> A8: Thanks for this suggestion. "Sufficiently diverse effect" means that the conditional distribution of sources given the auxiliary variable must vary sufficiently with the auxiliary variable and thus is more complex. This is often expressed in terms of the first-order and second-order derivatives of the conditional distribution; see the precise definition in Hyvärinen et al., (2019, Theorem 1). We will provide a detailed explanation in the revision.
>
> **Q9: "Do you have any explanations about the performance difference from two estimation methods in Figure 1?"**
>
> A9: Thanks for the thoughtful question. A possible reason is that the decomposition-based method involves additional constraint (Eq. (8)) as compared to likelihood-based method (Eq. (9)). Therefore, the resulting optimization problem of  decomposition-based method might be harder to solve and contain suboptimal local solutions. We will include this explanation in Section 5 of the revision.
>
> **References:**
>
> A. Hyvärinen, H. Sasaki, and R. Turner. Nonlinear ICA using auxiliary variables and generalized contrastive learning. In International Conference on Artificial Intelligence and Statistics, 2019.

---

> > ### Author Response · Authors · 2023-08-17
> > **A Kind Request for Further Feedback**
> >
> > Thanks again for taking the time to review our work. We have carefully considered your comments and provided responses to them. Since the discussion period will end in a few days, we would like to kindly request for further feedback. Could you please check whether the responses properly addressed your concern? Thank you very much.

---

> > ### Author Response · Authors · 2023-08-19
> > **Looking forward to your kind feedback**
> >
> > Dear Reviewer 6AWD,
> >
> > We are writing to kindly let you know that we have been eagerly waiting for your feedback on our rebuttal, despite your busy schedule.  Since the discussion period will end on Monday, we hope for the opportunity to respond to your further comments or questions, if there are any.  Any feedback would be appreciated.
> >
> > Thanks once again,
> >
> > Authors of #8190

---

> > > ### Author Response · Authors · 2023-08-21
> > > **A Kind Request for Further Feedback**
> > >
> > > Dear Reviewer 6AWD,
> > >
> > > We apologize for sending multiple reminders. Since the discussion period will end in two hours, we are very eager to get your feedback on our response. We understand that you are very busy. We would highly appreciate it if you could take into account our point-by-point response when updating the rating and having discussion with AC and other reviewers.
> > >
> > > Thanks for your time,
> > >
> > > Authors of #8190

---

> > > > ### Comment · Reviewer_6AWD · 2023-08-21
> > > >
> > > > Thank you for your responses and sorry for the late reply. I appreciate the authors kindly answered all of my questions. I choose to remain my score for now since there is no real data experiment result provided. I will raise my rating to 6 if you could provide a short summary (including figures and text summary) of the preliminary results on real data.

---

> > > > > ### Author Response · Authors · 2023-08-21
> > > > >
> > > > > Dear Reviewer 6AWD,
> > > > >
> > > > > Many thanks for your feedback.  We are writing to let you know that we are summarizing our results right now and will post them here ASAP.
> > > > >
> > > > > Best regards,
> > > > >
> > > > > Authors of #8190

---

> > > > > > ### Author Response · Authors · 2023-08-21
> > > > > >
> > > > > > Dear Reviewer 6AWD,
> > > > > >
> > > > > > Many thanks for your further comment and considering our response. Here, we provide a summary of the preliminary results on real-world audio datasets. Specifically, we have tested ten independent speech audios (containing English, Japanese, etc), each of these has a sample size of $160,000$. The length of each audio is $10$ seconds. Using SparseICA (our method), we achieved an average MCC of $0.8788$ with a standard deviation of $0.12489$. Meanwhile, FastICA recorded an average MCC of $0.74211$ with a standard deviation of $0.09991$. Because according to the instructions by PC, we are not allowed to share any link in our response, **we have additionally sent the link of the result plot (violin plot) to the AC privately and let the AC decide whether the link can be shared**.  Hope you can see it.
> > > > > >
> > > > > > Please let us know if you have any further questions.  Hope for opportunities to respond to them.
> > > > > >
> > > > > >
> > > > > > With best regards,
> > > > > >
> > > > > > Authors of #8190

---

> > > > > > > ### Comment · Reviewer_6AWD · 2023-08-21
> > > > > > >
> > > > > > > Thank you for your fast response! It is promising to see SparseICA did perform well on real data. I have updated my score to 6. Please do include real data results in the revised manuscript.

---

> > > > > > > > ### Author Response · Authors · 2023-08-21
> > > > > > > >
> > > > > > > > Many thanks for taking into account the response and experiments. We will incorporate the suggestions and results in the revised manuscript.

---

### Official Review · Reviewer_ouDH · 2023-07-11

**Soundness:** 3 good
**Presentation:** 3 good
**Contribution:** 3 good
**Rating:** 4
**Confidence:** 3

**Summary:**

This paper provides theorems under which the mixing matrix of linear ICA with Gaussian sources can be identified. While Gaussian sources are known to be unidentifiable in the classical ica theory, identifiability is possible if certain sparse structure is assumed for the mixing matrix as was initailly shown in Zheng. The assumptions in Zheng were restrictive however. This work expands the work of Zheng to provide more general conditions, namely:
- assumption of structural provided is much less restrictive on the sparsity pattern
- necessity of this assumption is proven (under the problem considered here)
- connection of this approach to causal discovery is shown
- the theorems lead to new estimation methods based on second-order statistics that allow ICA to be applied on Gaussian sources


**Strengths:**

Strengths:
- the paper is in general well written and of good quality
- the assumptions on sparsity are more much more reasonable than in previous works and should allow for future works in this area. To this end theorems 2 and 3 provide relevant and useful comparison
- theoretical assumptions are also often nicely illustrated with examples
- some nice approach of transforming the theorems into algorithms in a justified manner (e.g. framing the search space of A; theorem 6)
- connection between ica and causal discovery is well known and this paper further contributes to that similarity
- addition of a necessary condition is a nice result and helps to understand the limits of this approach




**Weaknesses:**

Weaknesses:

The biggest conceptual problem I have is that the identifiability here is framed in terms of an optimization problem where the ground-truth covariance matrix is assumed. I find this idea hard to follow since identifiability should be a property of the data model. Typically the process is to assume that we have $\log p(x; \theta) = \log p(x; \hat{\theta})$ and then show that this implies $\theta \sim \hat{\theta}$ -- this approach makes sense as the starting point, equality of likelihoods, can be justified on the basis of MLE and its guarantees (at least in theory). Here instead one seems to start with the assumption that $AA' = \hat{A}\hat{A}'$. But the justification for this starting point is missing -- you should write it out fully. Also, in practice the empirical covariance matrix is used (eq. 8) but what are the guarantees here? I'm quite happy to revise my score upwards once this, and below, are discussed.

I think the practical motivations for this paper are lacking and weak, perhaps partly due to a bit of unclear writing. For example, the authors write that "many biological traits ... are often normally distributed" to justify the importance of doing ica on Gaussian latents. But those listed biological signals are *not* typically latent, but often observed (after noise) and can usually be handled by standard ICA as there is no problem of Gaussian observations. Of course, it is possible to have latent biological signals too but that should be explained more carefully. It should be stressed however that the work still has important contribution in terms of fundamental research -- practical appeal does not always need to be obvious, so this is not necessarily a big problem. Authors also write that one possible situation in which their theorems are more useful than previous works, is when "an observed variable serves as a root cause". This may be possible but goes against the typical idea of ICA where observations are all assumed mixtures of independent latents. I feel this is however more easily understood in causal framwork. This leads to my next point...

I find in general that ideologically this work is in some sense difficult to place as it feels like the ideas are more relevant to causal discovery yet it's framed as an ICA paper. But the assumptions on sparsity can still be quite restrictive and a substantially sparse mixing matrix, again, goes against the whole concept of ICA in some sense -- we are not really solving the mixing matrix problem if the mixing that happens is quite limited. Despite the examples given, it is still hard to understand exactly how restrictive sparsity assumptions are *in practice*.

I find the estimation algorithm of the decomposition-based method in general a bit difficult to justify; optimally you would use an $l_o$ regularizer as this would agree with theory but that seems difficult to optimize. The likelihood-based method seems much more principled and I wonder if this paper could have been better framed around likelihood-based methodology in general.

I'm surprised by the poor experimental results -- typically linear ICA achieves 0.99 correlation to the ground-truths eg. with FastICA so I would have expected similar results here. There are also no experiments on real data (not necessarily a big problem due to the breadth of theory here).

**Questions:**

More concerns and things I'd expect revised for me to improve my score:

line 16-17: the examples+reference here are pretty much straight form Hyvärinen (Independent Component Analysis: Algorithms and Applications, section 7). Perhaps with a bit more originality and effort some more varied, newer, examples or at least references could be provided?

"distribution of Gaussian sources, the sparsity of the mixing matrix undergoes noticeable changes." Could you clarify what is meant here?

Please include "links" to relevant proofs in the text. At the moment one has to wonder in the appendix to see whether the proofs exist.

329: "...based method (Eq. (8) or Eq. (9)) on data where both Assumptions 1 and 2 do not hold" The grammar could be polished as now it could mean "where assumption 1 and 2 do not hold at the same time"  or that "neither assumptions 1 and 2 hold"

**Limitations:**

Authors do indeed point out the limited applications to real world data. Authors also admit that " Since the true generating process of real-world data is inaccessible, it is challenging to quantitatively  evaluate the applicability of these sparsity assumptions."

however there is no discussion on why the experimental results are not as good as one would expect, this would be welcome.

---

> ### Author Rebuttal · Authors · 2023-08-09
>
> We greatly appreciate the reviewer's constructive comments, many of which will help improve the clarity of our paper. We have tried to address all the concerns in the following.
>
> **Q1: Justification to start with $AA'=\tilde{A}\tilde{A}'$ is missing.**
>
> A1: We sincerely appreciate this insightful comment, which helps improve the clarity of our theoretical results. We agree that the typical starting point of identifiability proof is "equality of likelihoods". In fact, this is exactly the tool used in our proof of likelihood method. In the large sample limit (as is the case for typical justification of MLE), one can show that $L(\mathbf{A};\bar{\mathbf{\Sigma}})=L(\tilde{\mathbf{A}};\bar{\mathbf{\Sigma}})$ implies $\mathbf{A}\mathbf{A}^\top=\tilde{\mathbf{A}}\tilde{\mathbf{A}}^\top$ (see Eq (9) for exact form of likelihood $L(\cdot)$).
>
> In Theorems 1 and 6, we start with $\mathbf{A}\mathbf{A}^\top=\tilde{\mathbf{A}}\tilde{\mathbf{A}}^\top$ because this is the essential assumption used by the proof. This leaves the door open for different ways to achieve $\mathbf{A}\mathbf{A}^\top=\tilde{\mathbf{A}}\tilde{\mathbf{A}}^\top$, i.e., via decomposition or likelihood method, both of which are correct in the large sample limit. We will discuss this further in the revision.
>
> **Q2: Guarantees for empirical covariance matrix.**
>
> A2: See our response to Q1 in the general response.
>
> **Q3: lack of practical motivations and "it is possible to have latent biological signals too but that should be explained more carefully".**
>
> A3: Thanks for the suggestion. In light of it, we will modify the sentence
> to emphasize that the real-world usage scenarios center on potential latent Gaussian sources. We will also include additional examples besides biology. For instance, the thermal noises in electronic circuits typically adhere to Gaussian distributions (Ott, 1988), of which their mixtures posing challenges to traditional separation techniques.
>
> **Q4: "an observed variable serves as a root cause" and "goes against the typical idea of ICA".**
>
> A4: Thank you for pointing this out. In light of your comment, we will remove that related part to avoid possible misunderstanding.
>
> **Q5: "assumptions on sparsity can still be quite restrictive" and "it is still hard to understand exactly how restrictive sparsity assumptions are".**
>
> A5: Thanks a lot for this thoughtful comment. We completely agree with you that our assumption may be violated in certain situations, and we do not expect our theory to apply to all scenarios. However, given that the problem has clear practical implications and that the sparsity assumptions are expected to hold true for certain mixing matrices, it seems essential to start this line of research, and to weaken the sparsity assumptions as much as possible. This extends the applicability of these approaches to cover more general mixing matrices. We also hope that this work will inspire alternative results to make gaussian sources identifiable.
>
> It is worth noting that sparsity assumptions are particularly relevant when observations are influenced by sources in a "simple" manner, as also discussed in [33]. For instance, ecological, gene-regulatory, and metabolic systems in biology often exhibit sparse interactions (Busiello et al., 2017). Similarly, in physics, complex observed phenomena may often be governed by a relatively small set of fundamental laws, exemplified by Einstein's theory of special relativity (Einstein, 1905). In our revised paper, we will delve into these aspects further, providing a more comprehensive perspective on the practical implications of our theory.
>
> **Q6: Frame the paper around likelihood-based methodology.**
>
> A6: Thanks for this constructive suggestion. We agree that optimally we would use an $\ell_0$ regularizer for decomposition-based method. This partly explains why likelihood method performs slightly better than decomposition method. Following your suggestion, we will restructure Section 4.2 to place more emphasis on the likelihood method.
>
> **Q7: "no experiments on real data".**
>
> A7: See our response to Q2 in the general response.
>
> **Q8: Provide more original, varied, newer examples/references.**
>
> A8: Beside examples in Lines 16-17, we will include biology (Teschendorff et al., 2007; Biton et al., 2014), astronomy (Nuzillard et al., 2000; Akutsu et al., 2020), and earth science (Kaplan, 2003; Moulin et al., 2022) in the revision.
>
> **Q9: Clarify "sparsity of the mixing matrix undergoes noticeable changes".**
>
> A9: This means that, after rotation of the mixing matrix, the support of the mixing matrix may be changed, leading to a denser mixing matrix, although the resulting distribution of the observed variables remains unchanged. Similar intuition is explained in Lines 172-174. We will clarify this further in the revision.
>
> **Q10: Include links to proofs.**
>
> A10: We will include the links in main texts to guide readers to the relevant proofs in the Appendix sections.
>
> **Q11: Polish the grammar of "329: ...based method ... do not hold".**
>
> A11: We will modify the phrase to "where neither Assumption 1 nor Assumption 2 holds".
>
> **Q12: "typically linear ICA achieves 0.99 correlation" and "no discussion on why the experimental results are not as good as one would expect".**
>
> A12: Thanks for bringing up this question and your insightful observation. Since the experiments involve Gaussian sources, FastICA does not perform well because it is based on non-Gaussianity. For our method that can handle Gaussian sources, the optimization may return suboptimal local solutions, so the experimental results are not as good as one would expect (despite still outperforming FastICA). This further demonstrates the difficulties posed by Gaussian sources due to rotational invariance, and indicates that further research could enhance the performance. We will discuss this in the revision.

---

> > ### Author Response · Authors · 2023-08-15
> > **A Kind Request for Further Feedback**
> >
> > Thanks again for your time and comments. We have provided responses to your comments. Would you mind checking whether they properly addressed your concerns, or if you have further comments? Your feedback would be appreciated.

---

> > ### Comment · Reviewer_ouDH · 2023-08-17
> >
> > I have re-read the paper and the other reviews, comments, rebuttals etc and have reconsidered my opinion.
> >
> > While my opinion of the paper is improved I am still also not fully convinced whether the paper quite merits acceptance. The theoretical work is very good and it does indeed relax conditions compared to previous work, but it I don't find it significant enough on its own as the conditions are still quite strong and restrictive and build quite clearly on previous ideas. I would thus expect more strong empirical results to really show me that "here we have a real-world problem and as you can see Zheng's approach fails, as do typical ICA approaches, but our model does much better". You do mention you are running some tests on Richard et al. experiments but I find those impossible to judge, as I have not seen them.  **In conclusion: optimally, I would recommend the authors resubmit with more convincing empirical, real-world, results as I think it would make the paper a lot stronger but I will still engage in further discussion with other reviewers as to me this paper is very much on the threshold.**
> >
> >
> >
> >
> >
> >
> >
> > p.s. minor thing:
> > **Q1** I think you missed my point here (and perhaps it was unclear on my part). I understand well where these are coming from e.g. as you say . *"In the large sample limit (as is the case for typical justification of MLE), one can show that $L(A)=L(\hat{A})$ implies $AA^T = \hat{A}^T \hat{A}$."* All I am saying you should probably say this more explicitly in the main text. This would make the connection to typical identifiability theorems explicit to readers.

---

> > > ### Author Response · Authors · 2023-08-17
> > >
> > > We thank the reviewer for reading our response and for the further comment. We are glad that your opinion of the paper is improved, and you acknowledge that the theoretical work is very good.
> > >
> > > Regarding "build quite clearly on previous ideas", while the tool of sparsity is inspired by Zheng et al. (2022), we note that the technical development is entirely different. Our theory/proof leverages the distributional constraints of covariance matrices and effects of support rotations, thereby allowing us to relax the conditions. Such development and techniques were not seen in Zheng et al. (2022) (as well as the typical literature of ICA).
> > >
> > > Regarding "the conditions are still quite strong and restrictive", **this is precisely the motivation of our work**. We view it as our duty as researchers to progressively relax these conditions, thereby extending the applicability of the theory to cover more general mixing matrices. Due to the rotational invariance of the Gaussian distribution, ICA with Gaussian sources is a challenging ill-posed problem, and thus some assumptions are inevitably needed. As you acknowledged, we further showed that some of the assumptions are provably necessary, which shed light on the limit of these approaches.
> > >
> > > We admit that our work is primarily centered around theory. While we agree that the potential application of our theory in various real-world tasks would be exciting, we note that our theory can only be rigorously validated by ablation studies (via simulated data), which we have done in Section 5. This is because, as you also noticed, it is impossible to ensure that the unknown ground-truth data generating process satisfies some assumptions or not. Therefore, real-world experiments cannot play the role of validating the theory.
> > >
> > > Furthermore, it is worth noting that NeurIPS has traditionally published theory-focused papers if they offer relevant results and use nontrivial methods. We believe that both are the case.
> > >
> > > Regarding Q1, thanks for the clarification and we will make this more explicit in the main text, following your suggestion.

---

> > > > ### Comment · Reviewer_ouDH · 2023-08-20
> > > >
> > > > Just wanted to acknowledge that I've considered these comments but I hold my opinion of above that to me this is a borderline paper with important and good quality, theoretical advances but I also feel that they are not novel enough + suffer from strong restrictions (e.g. assumption 1). I don't buy the authors point that strong and restrictive conditions is only a good thing -- I think they show also that this approach is likely not practical. Yes simulated experiments are good at validating theory (and they could have been more extensive) I feel this is not strong enough of a 'theory paper' but would have needed experimental advances to show that it has practical value. Large number of previous, theoretical, NICA papers have managed to show the value of their algorithms on different types of real data. The more I think about the authors answers, the more I also worry about other issues such as how will these results hold when we want to estimate small number independent components for large data (observed dim >> latent dim) which is needed in a lot of situations.

---

> > > > > ### Author Response · Authors · 2023-08-20
> > > > >
> > > > > Thanks a lot for the further response. Regarding "I don't buy the authors point that strong and restrictive conditions is only a good thing", it seems that there might be some misunderstandings w.r.t. to our previous response. We intended to say that, as you mentioned, since the conditions are quite strong and restrictive in this line of approaches (e.g. Zheng et al. (2022)), *“we view it as our duty as researchers to progressively **relax** these conditions”* to *“cover more general mixing matrices”*, which is the motivation of this work and has been emphasized throughout the paper. Specifically, some major conditions in this setting have been relaxed to necessary (weakest possible) ones. We did **not** say that strong and restrictive conditions are good things at all, and tried our best to make the existing ones as weak as possible. Regarding real data, we did not provide the final/full results in the original rebuttal due to the time constraint of the rebuttal period, and will report them in the revision. Thanks again. We are looking forward to knowing whether that has been clarified or not from your perspective.

---

> > > > > > ### Comment · Reviewer_ouDH · 2023-08-20
> > > > > >
> > > > > > I apologise for my misunderstanding. That makes sense thanks. I will keep my score as is since I don't really know how to take into account the work on real data. But I will make the case to the AC that if we can take such things (unseen work) into account then I would raise the score to borderline accept.

---

> > > > > > > ### Author Response · Authors · 2023-08-20
> > > > > > >
> > > > > > > Thanks for taking into account the response and for discussing with the AC regarding updating the score.

---

> > > > > > > > ### Author Response · Authors · 2023-08-21
> > > > > > > >
> > > > > > > > Dear Reviewer ouDH,
> > > > > > > >
> > > > > > > > Thanks again for your time and efforts. In light of the comment regarding real data,  we provide a summary of the preliminary results on real-world audio datasets. Specifically, we have tested ten independent speech audios (containing English, Japanese, etc), each of these has a sample size of $160,000$. The length of each audio is $10$ seconds. Using SparseICA (our method), we achieved an average MCC of $0.8788$ with a standard deviation of $0.12489$. Meanwhile, FastICA recorded an average MCC of $0.74211$ with a standard deviation of $0.09991$. Because according to the instructions by PC, we are not allowed to share any link in our response, **we have additionally sent the link of the result plot (violin plot) to the AC privately and let the AC decide whether the link can be shared**.  Hope you can see it. Please let us know if you have any further questions.
> > > > > > > >
> > > > > > > > With best regards,
> > > > > > > >
> > > > > > > > Authors of #8190

---

> > > > > > > > > ### Comment · Reviewer_ouDH · 2023-08-21
> > > > > > > > >
> > > > > > > > > Thank you -- Dear AC, would you be okay with sharing that figure. I believe it would be valuable as it would likely influence my final score of the paper. I think the authors have done great job trying to provide it in time so it would feel justified to share it.

---

### Official Review · Reviewer_vUDk · 2023-07-16

**Soundness:** 3 good
**Presentation:** 3 good
**Contribution:** 3 good
**Rating:** 7
**Confidence:** 3

**Summary:**

This paper considers the problem of estimating a mixing matrix $A^*$, given measuremnts $x = A^* s$. The authors show that under Gaussian $s$, some assumptions on $A$, and using infinite $x$, the sparsest $A$ that satisfies $AA^{T} = A^* A^{* T}$ recovers $A^*$ upto permutation and sign of the columns.

The authors also show connections to causal discovery in the case of structural equation models.

Since the sparse norm is non-convex, the authors propose a convex relaxation using the $\ell_1$-norm that can be run in practice.

The sparsity assumptions on $A$ induce polynomial constraints $H(A)$ on the elements of the matrix $\Sigma = AA^T$. These constraints are typically hard to compute, but the authors define an auxilliary function $h(A) = \text{tr} ( \sum_{k=2}^{n} off(A) \odot off(A) )$, where $off(A)$ is the off-diagonal version of $A$, and $\odot$ is the Hadamard product. The constaint set now becomes $h(A) = 0$, and the authors show that convex solvers can work in this setting.


Minor:
Using $H$ for the set of hard constraints and $h$ for the trace of the defined matrix is slightly confusing, please change it.

**Strengths:**

- The authors show that their assumptions on the matrix are strictly weaker than existing approaches (Section 3.3)
- The authors show connections to causal discovery, which I thought were interesting.
- Existing approaches cannot handle Gaussian sources $s$, as this induces rotational symmetry on $AA^T$, and hence we can only recover $A$ upto rotation. In contrast, the current work can recover it upto permutation and sign of the matrix.
- The authors are able to show that the convex relaxation in Section 4 recovers $A$ in the asymptotic limit of infinite samples of $x$.
- Experimental results are nice.


**Weaknesses:**

- The identifiability results are for infinite samples of $x$.

- No runtime guarantees

- The authors should spend more time commenting on the technical quality of the results. In some sense, the extension from the non-convex program to the convex relaxation is ``expected''. What new technical tools were required over traditional sparsity?

- Experiments are a little simplistic, but that's fine for a theory paper.

**Questions:**

- I'm not an expert on causal discovery. Are the connections between ICA and identifiability of structural equations established results? Relatedly, can you explain your contributions over existing work.

- The authors should spend more time commenting on the technical quality of the results. In some sense, the extension from the non-convex program to the convex relaxation is ``expected''. What new technical tools were required over traditional sparsity? Without commenting on this, it would seem that choosing the right assumptions (1-3) are the key for this paper.

---

> ### Author Rebuttal · Authors · 2023-08-09
>
> We thank the reviewer for the positive feedback and time devoted to our work. Below we give a point-by-point response to the comments.
>
> **Q1: "Using $H$ for the set of hard constraints and $h$ for the trace of the defined matrix is slightly confusing, please change it".**
>
> A1: Thanks for this suggestion which helps improve the notations. We will change $h(\mathbf{A})$ to other notation such as $g(\mathbf{A})$ in the revision.
>
> **Q2: "The identifiability results are for infinite samples".**
>
> A2: See our response to Q1 in the general response.
>
> **Q3: "No runtime guarantees".**
>
> A3: Thanks for pointing this out. This is an excellent point. Here, we briefly discuss the overall runtime/complexity of the estimation method. For instance, considering the likelihood-based method  (i.e., Algorithm 2) with L-BFGS, each inner iteration of the quadratic penalty method (corresponding to the L-BFGS run) has a computational complexity of $O(m^2 n^2 + m^3 + m n^2 t )$, where $m\ll n^2$ is the memory size of L-BFGS and $t$ is the number of iterations of L-BFGS. Typically, we have $t=250$ for each L-BFGS run, and $125$ iterations for the quadratic penalty method. In practice, for the experiments in Figure 1, the average runtime of the likelihood-based method is roughly $2$ minutes on CPUs. (It is worth noting that the runtime can be significantly shortened with GPU acceleration.) We will provide this discussion with further explanation in the revision.
>
> **Q4: "Are the connections between ICA and identifiability of structural equations established results? Relatedly, can you explain your contributions over existing work."**
>
> A4: Thanks for this insightful question, which helps make our contributions clearer. As acknowledged by Reviewer ouDH and discussed in Lines 201-206, the connection between ICA and identifiability of structural equations has been established based on non-Gaussianity. One of our contributions is to extend the connection between ICA and structural equations to the case of Gaussianity (i.e., second-order statistics), which further bridges the gap between these two fields. Specifically, we provide an analogy of score-based causal discovery method (based on sparsity and second-order statistics) in ICA, which is rather different from the known connection based on non-Gaussianity. We will provide a detailed discussion to make our contributions for this part clearer in the revision.
>
> **Q5: "What new technical tools were required over traditional sparsity?"**
>
> A5: We sincerely appreciate this thoughtful question. Apart from traditional sparsity, we completely agree with the reviewer that choosing the right assumptions 1 to 3 are important for the identifiability/identification. Furthermore, the characterization (i.e., Proposition 4) of matrices $\mathbf{A}$ satisfying Assumption 2 (see Eq. (3)) is also a key technical tool of the proposed identification method, because it allows us to formulate the problem as a continuous constrained optimization problem. We will include this discussion with further explanation in the final version.
>
> **Q6: "Experiments are a little simplistic".**
>
> A6: See our response to Q2 in the general response.

---

### Official Review · Reviewer_zk7j · 2023-07-21

**Soundness:** 3 good
**Presentation:** 3 good
**Contribution:** 3 good
**Rating:** 6
**Confidence:** 3

**Summary:**

In this paper, the authors develop new identifiability conditions for ICA, based on sparsity constraints. These assumptions, in particular, do not require the sources to be non-Gaussian, as is usually the case in ICA.
Mostly, the authors take inspiration in previous work by Zheng et al, and weaken their main condition. Some assumption are shown to be necessary.
As is usual with sparsity-based analyses, the resulting optimization problem is quite non-trivial, here the main difficulty is to explore the space of directed adjacency matrices whose corresponding graph is a DAG. The authors provide some discussion and implementation procedures about this.
The proposed method is evaluated against FastICA on synthetic data. A (quick) ablation study is realized to examine the effect of the assumptions not being satisfied.

**Strengths:**

- the contribution is interesting for the long-standing problem of non-Gaussian ICA. The authors make a commendable effort to discuss previous results, their main inspiration, and the strengthening of the results/weakening of assumptions

- the paper is well-written and pedagogical, with examples and discussion scattered throughout about the assumptions and the results

- some aspects of the paper are quite developed (relation with causal discovery, implementation, necessity of the assumptions...) without sacrificing the clarity of the writing despite the space constraints

**Weaknesses:**

- the only assumption that less discussed than the other ones, in relation with previous work or its "realistic aspect" with respect to real data, seems to also be the "main" one, Assumption 2. The focus on hard constraints is theoretically understandable, but how realistic is it ? Also, the triple-presentation of the optimization problem is a bit redundant, the end problem being "just" optimizing among the space of DAGs

- the paper focuses on exact identifiability, when the true covariance matrix is "available". This can be confusing for people not expert on ICA, can the effect of sample size and covariance estimation be discussed ? Does the resulting noise has an effect on "admissible" sparsity level (empirically, let's say), and for now the sparsity level $\| A\|_0$ does not seem to be constrained in any way (beyond assumption 1 and 2)

- there is no discussion on computational complexity, it seems that the constraint $h(A)=0$ is very costly as it involves computing the $n$th power of an $n \times n$ matrix, at each iteration. Is there any simplification over brute force ?

- no evaluation on real data. When the assumptions are only approximately satisfied, and there is strong noise on the covariance, it is not clear what is the performance of the proposed method

**Questions:**

See "weaknesses" above

Minor questions/typos:

- l22 : "Kurtois"
- l 95 : "statitsic"
- Assumption 3 is a bit unclear at first read (it seems like a definition), only one sense of the "if and only if" is important, perhaps it is possible to reformulate
- what is the exact sense of "asymptotically" in theorem 7 ? (see also question/comment about sample size, which is not discussed in the paper)

**Limitations:**

The authors have discussed some aspects extensively, eg Assumption 1 with respect to previous work, other are a bit scarce, eg Assumption 2. Also, the computational complexity is not discussed, and there is no experiments on real data.

---

> ### Author Rebuttal · Authors · 2023-08-09
>
> We sincerely thank the reviewer for the time devoted and the thoughtful comments. Please find the response to your comments and questions below.
>
> **Q1: Lack of discussion about Assumption 2 and how realistic it is.**
>
> A1: We sincerely appreciate this very helpful point. Since Assumption 2 allows independent column and row permutations, we believe that it is rather mild, especially for sparse mixing matrices. In other words, if the influences from sources to observed variables are sparse, then it is likely for Assumption 2 to hold. Such notion of sparse influences tend to apply when the observations are influenced by the sources in a "simple" manner, e.g., in several biological (Busiello et al., 2017) and physical (Einstein, 1905) systems. Furthermore, sparse influences also serve as the fundamental principle for various works in ICA (Zheng et al., 2022; Lachapelle et al., 2022), as well as other fields like causality (Spirtes et al., 2001; Raskutti et al., 2018). See also our response to Q5 for Reviewer ouDH. We will provide this discussion with further elaboration in the revision.
>
> **Q2: "the triple-presentation of the optimization problem is a bit redundant".**
>
> A2: Thanks for this suggestion. We will update the presentation of the optimization problem in the revision to make it concise.
>
> **Q3: "Can the effect of sample size and covariance estimation be discussed?"**
>
> A3: See our response to Q1 in the general response.
>
> **Q4: "Does the resulting noise has an effect on 'admissible' sparsity level".**
>
> A4: We sincerely appreciate this insightful question. In the current theoretical results, the resulting noise does not have an effect on "admissible" sparsity level. That is, as the reviewer nicely noted, the sparsity level $\|\mathbf{A}\|_0$ is not constrained in any way beyond Assumptions 1 and 2. See also our reply to Q1 in the general response.
>
> **Q5: Computational complexity and "is there any simplification over brute force".**
>
> A5: Thanks for raising this excellent question which helps improve the clarity of the method. As the reviewer nicely mentioned, a straightforward and brute-force approach to compute the constraint term $h(\mathbf{A})=\operatorname{tr}\left(\sum_{k=2}^n (\operatorname{off}(\mathbf{A})\odot \operatorname{off}(\mathbf{A}))^k\right)$ is to compute each matrix power in $h(\mathbf{A})$ and then sum their traces up; this approach requires $O(n)$ matrix multiplications. In our implementation, we adopt a more efficient approach with computational complexity of $O(\log n)$ matrix multiplications, inspired by Zhang et al. (2020). The rough idea is to perform exponentiation by squaring (i.e., a general procedure similar to binary search) and recursively compute the term $h(\mathbf{A})$. We will provide a detailed description of this procedure and its computational complexity in the final version.
>
> **Q6: "no evaluation on real data".**
>
> A6: See our response to Q2 in the general response.
>
> **Q7: Typos.**
>
> A7: Thanks for your careful reading. We will carefully proofread the paper to fix the typos in the revised version.
>
> **Q8: "Assumption 3 is a bit unclear at first read" and "only one sense of the 'if and only if' is important".**
>
> A8: Thanks for your constructive suggestion which helps improve the clarity of the assumption. We completely agree with the reviewer that only one sense of the "if and only if" is important. In light of your comment, we will reformulate Assumption 3 as follows:
> > **Assumption 3 (Faithfulness).** The resulting covariance matrix $\mathbf{\Sigma}=\mathbf{A}\mathbf{A}^\top$ of mixing matrix $\mathbf{A}$ satisfies a hard constraint $\kappa$ only if $\kappa\in H(\boldsymbol{\xi}_\mathbf{A})$.
>
> **Q9: "What is the exact sense of 'asymptotically' in theorem 7?"**
>
> A9: Thanks for asking this question. By "asymptotically", we intend to mean "in the large sample limit". That is, an infinite number of samples are given so that we have access to the true covariance matrix. We will update the term "asymptotically" to "in the large sample limit" in the revision to avoid possible confusion.
>
> **References:**
>
> D. M. Busiello, S. Suweis, J. Hidalgo, and A. Maritan. Explorability and the origin of network sparsity in living systems. Scientific reports, 7(1):1–8, 2017.
>
> A. Einstein. Does the inertia of a body depend upon its energy-content. Annalen der Physik, 18(13): 639–641, 1905.
>
> Y. Zheng, I. Ng, and K. Zhang. On the identifiability of nonlinear ICA: Sparsity and beyond. In Advances in Neural Information Processing Systems, 2022.
>
> S. Lachapelle, P. R. López, Y. Sharma, K. Everett, R. L. Priol, A. Lacoste, and S. Lacoste-Julien. Disentanglement via mechanism sparsity regularization: A new principle for nonlinear ICA. Conference on Causal Learning and Reasoning, 2022.
>
> P. Spirtes, C. Glymour, and R. Scheines. Causation, Prediction, and Search. MIT press, 2nd edition, 2001.
>
> G. Raskutti and C. Uhler. Learning directed acyclic graph models based on sparsest permutations. Stat, 7(1):e183, 2018.
>
> Z. Zhang, I. Ng, D. Gong, Y. Liu, E. M. Abbasnejad, M. Gong, K. Zhang, and J. Q. Shi. Truncated matrix power iteration for differentiable DAG learning. In Advances in Neural Information Processing Systems, 2022.

---

> > ### Comment · Reviewer_zk7j · 2023-08-16
> > **Response to rebuttal**
> >
> > I thank the authors for their careful rebuttal, which answers many of my concern. I will keep my score as is, but argue for accept.

---

> > > ### Author Response · Authors · 2023-08-16
> > > **Thanks**
> > >
> > > We are grateful to the reviewer for reading our response and for the acknowledgement. We will incorporate the suggestions in the revision. Many thanks.

---

### Author Rebuttal · Authors · 2023-08-09

We thank all of the reviewers for the valuable feedback and time devoted to reviewing our work. We are encouraged that they found our work to be interesting (zk7j, vUDk), developed (zk7j), and useful (6AWD). We are grateful that the reviewers appreciate our theoretical (zk7j, vUDk, ouDH, 6AWD) and methodological (vUDk, ouDH, 6AWD) contributions; specifically, Reviewer ouDH believes that our "assumptions on sparsity are more much more reasonable than in previous works and should allow for future works in this area". Reviewers zk7j, ouDH, and 6AWD also appreciate that the theoretical assumptions/results are presented in a clear way, often illustrated with explanations and examples.

We take the opportunity to clarify below common questions raised by the reviewers. We also provide individual responses to address the comments of each reviewer.

**Q1: Identifiability for finite samples.**

A1: We greatly appreciate this insightful comment. It is worth noting that ICA with Gaussian sources is a challenging ill-posed problem due to the rotational invariance of the Gaussian distribution. Therefore, establishing identifiability results for infinite samples is itself a fundamental problem, and this type of "possibility" identifiability results are needed before one is able to establish guarantees on finite samples.

Given our identifiability result on infinite samples, it will become clear that the problem is solvable under appropriate assumptions. The next step is then to extend this result for finite samples. The empirical studies in Appendix E demonstrate the effectiveness of our method for finite samples, thereby showcasing the potential to extend our identifiability result from infinite to finite samples. We will provide this discussion in the revision.

**Q2: Experiments on real data.**

A2: As acknowledged by Reviewers vUDk and ouDH, the lack of experiments on real data is not necessarily an issue "due to the breadth of theory here". At the same time, in light of the comment, we will incorporate an additional experiment on a real dataset with fMRI data. Due to the time constraint, we are conducting preliminary experiments using the dataset considered by Richard et al. (2021), by adapting the multi-subject data to suit our specific context. We find that the preliminary results align with the observations reported in Section 5. We will include experimental details and final results in the revision.

**References**

H. Richard, P. Ablin, B. Thirion, A. Gramfort, and A. Hyvarinen. Shared independent component analysis for multi-subject neuroimaging. In Advances in Neural Information Processing Systems, 2021.

---

> ### Author Response · Authors · 2023-08-21
> **Preliminary Results on Real-World Dataset**
>
> We are very grateful for all the insightful discussions. Due to the time limit and the specific requirement of Reviewer 6AWD and Reviewer ouHD, here we provide a summary of the preliminary results on real-world audio datasets. Specifically, we have tested ten independent speech audios (containing English, Japanese, etc), each of these has a sample size of $160,000$. The length of each audio is $10$ seconds. Using SparseICA (our method), we achieved an average MCC of $0.8788$ with a standard deviation of $0.12489$. Meanwhile, FastICA recorded an average MCC of $0.74211$ with a standard deviation of $0.09991$. Because according to the instructions by PC, we are not allowed to share any link in our response, **we have additionally sent the link of the result plot (violin plot) to the AC privately and let the AC decide whether the link can be shared**. Please kindly let us know if there are any further questions.

---

### Decision · Program_Chairs · 2023-09-21

**Decision:**

Accept (poster)

**Comment:**

This paper proposes a new theory for the identifiability of linear ICA when the sources are Gaussian. It presents new assumptions on the mixing matrix that allows recovery.

The proposed theory is sound, and the assumptions are weaker than those of previous works. The preliminary experiments were quite toy but this is not a problem from a theory paper, and it does not raise concerns regarding the soundness of this work.

All reviewers except reviewer ouDH advocate for acceptance. Initial concerns raised by rev. ouDH were addressed by the authors; then rev. ouDH pointed out the lack of real-world experiments, which the authors addressed in their last experiment.